# Breaking the $\log(1/\Delta_2)$ Barrier: Better Batched Best Arm Identification with Adaptive Grids

**Tianyuan Jin**[1,*], **Qin Zhang**[2,*], **Dongruo Zhou**[2,*]
[1]Department of Electrical and Computer Engineering, National University of Singapore
[2]Department of Computer Science, Indiana University Bloomington
tianyuan@nus.edu.sg, qzhangcs@iu.edu, dz13@iu.edu

## Abstract

We investigate the problem of batched best arm identification in multi-armed bandits, where we aim to identify the best arm from a set of $n$ arms while minimizing both the number of samples and batches. We introduce an algorithm that achieves near-optimal sample complexity and features an instance-sensitive batch complexity, which breaks the $\log(1/\Delta_2)$ [1] barrier. The main contribution of our algorithm is a novel sample allocation scheme that effectively balances exploration and exploitation for batch sizes. Experimental results indicate that our approach is more batch-efficient across various setups. We also extend this framework to the problem of batched best arm identification in linear bandits and achieve similar improvements.

## 1 Introduction

In online learning, it is common to process data in batches each with a fixed policy, as frequently policy changes may incur significant additional costs. For example, in clinical trials, patients are typically treated the same at a time, since every policy switch would trigger a separate approval process (Thompson, 1933; Robbins, 1952). In crowd-sourcing, it takes time for the crowd to answer questions, and thus a small number of rounds of interaction with the crowd is critical in time-sensitive applications (Kittur et al., 2008). Similar phenomena occur in compiler optimization (Ashouri et al., 2019), hardware placements (Mirhoseini et al., 2017), database optimization (Krishnan et al., 2018), etc. Therefore, the design of batch-efficient learning algorithms is of central importance in online learning. In this paper, we study the following problem.

**Best Arm Identification in Multi-Armed Bandits.** In multi-armed bandits (MAB), we have a set of $n$ arms, each associated with a reward distribution $\mathcal{N}(\mu_i, 1)$, where $\mu_i$ is an unknown mean. Pulling the $i$-th arm gives a reward sampled from its reward distribution. In the problem of best arm identification in MAB (BAI-M), the goal is the identify the arm with the highest mean $\mu_* = \max_{i \in [n]} \mu_i$ with success probability $(1 - \delta)$ for a given $\delta$ using the minimum number of pulls/samples.[2] We assume that there exists a unique best arm $\mu_*$ among the $n$ arms.

**Batched Learning.** In the batched model, learning progresses in rounds, where the set of arms to pull must be decided at the beginning of each round. Let $t_1(= 1), t_2, \ldots, t_M$ denote the starting time steps of the $M$ batches. For convenience, we define $t_{M+1} = T + 1$, where $T$ is the final time step. The $i$-th batch spans from time steps $t_i$ to $(t_{i+1} - 1)$, and we refer to $(t_{i+1} - t_i)$ as the size of the $i$-th batch. There are two variations of this model: the first is the *fixed grid*, where $t_1, \ldots, t_M$ are determined at the at the very beginning of the learning process, and the second is the *adaptive grid*, where $t_i$ can be chosen based on the rewards observed up to time $(t_i - 1)$.

---

*Alphabetic author order

[1]$\Delta_2$ is the difference of means between the best arm and the second best arm.

[2]In this paper, we consider the *fixed-confidence* variant of BAI-M. The other variant is referred to as *fixed-budget*, where given a sample budget $T$, we want to identify the best arm with the smallest error probability.

In this paper, we focus on the adaptive grid setting and investigate the design of batched algorithms for BAI-M. We will also extend our results to best arm identification in linear bandits (BAI-L), with its formal definition deferred to Section 4.

**Sample and Batch Complexity.** For a batched learning algorithm, we define its *sample complexity* as the total number of arm pulls required to identify the best arm, and its *batch complexity* as the total number of batches needed to accomplish this task.

It is well-known that for BAI-M, the sample complexity of the algorithm can be made *instance-sensitive*. That is, the sample complexity can be written as a function of the input parameters, which can be much better than the minimax/instance-independent bound for many inputs. Let $I = \{\mu_i\}_{i=1}^n$ be an input instance for BAI-M. W.l.o.g., assume that $\mu_1 > \mu_2 \geq \ldots \geq \mu_n$. Let $\Delta_i = \mu_1 - \mu_i$ be the mean gap between the best arm and the $i$-th best arm. Several algorithms (Even-Dar et al., 2002; Even-Dar et al., 2006; Audibert et al., 2010; Kalyanakrishnan et al., 2012; Karnin et al., 2013; Jamieson et al., 2014; Carpentier & Locatelli, 2016; Chen et al., 2017) are able of achieving sample complexities on the order of $\tilde{O}(H_I)$, where '~' hides some logarithmic factors, and

$$H_I \triangleq \sum_{i=2}^n \frac{1}{\Delta_i^2} \tag{1.1}$$

is called the *instance sample complexity* of the input $I$. Among these algorithms, the *successive elimination (SE)* algorithm proposed in Even-Dar et al. (2002) can be naturally adapted to the fixed grid batched setting with a batch complexity of $\log(1/\Delta_2)$.

Preserving the optimal sample complexity $H_I$, the bound $\log(1/\Delta_2)$ has been proven as nearly minimax-optimal for the batch complexity of BAI-M, even in the adaptive grid setting (Tao et al., 2019).[3] However, this does not exclude the possibility of designing instance-sensitive batched algorithms that outperform the successive elimination algorithm for many inputs. In this paper, we try to address the following question:

*In the adaptive grid setting, can we design bandit algorithms for BAI-M that achieve nearly optimal sample complexity while breaking the $\log(1/\Delta_2)$ barrier for batch complexity?*

## 1.1 OUR CONTRIBUTIONS

In this paper, we answer the above question affirmatively. We propose new algorithms for batched best arm identification for both multi-armed bandits and linear bandits. Both algorithms achieve a smaller batch complexity for many inputs (and are never worse for all inputs) compared to the state-of-the-art algorithms (Even-Dar et al., 2002; Fiez et al., 2019a).

Our contributions are summarized in the followings: First, we propose an *instance-sensitive* quantity $R_I$ to capture the batch complexity of an instance $I$ in MAB.

**Definition 1.1** (Instance-sensitive batch complexity of MAB). *Set $\bar{L}_0 = 1$, $U_0 = \emptyset$, and $C = 15\sqrt{2}$. We recursively define for $r = 1, 2, \ldots$ the quantities*

$$\bar{L}_r = 4\bar{L}_{r-1} + \frac{1}{n - |U_{r-1}|} \sum_{j \in U_{r-1}} \frac{1}{\Delta_j^2}, \quad and \quad U_r = \left\{ j : \Delta_j \geq \frac{C}{\sqrt{\bar{L}_r}} \right\}. \tag{1.2}$$

*We stop when $U_r = [n] \setminus \{1\}$. Let $R_I$ be the value of $r$ when we stop.*[4]

Intuitively speaking, $\bar{L}_r$ represents the pull budget on each remaining arm in the $r$-th batch. Definition 1.1 introduces *adaptive grids* for batch sizes that are instance-dependent (as they depend on the gaps

---

[3]Tao et al. (2019) showed that to achieve $\tilde{O}(H_I)$ time complexity in the $O(1)$-agent collaborative learning model, $\Omega\left(\log \frac{1}{\Delta_2} / \log\log \frac{1}{\Delta_2}\right)$ rounds of communication is necessary. This lower bound can be straightforwardly translated to the batched learning model, proving an $\Omega\left(\log \frac{1}{\Delta_2} / \log\log \frac{1}{\Delta_2}\right)$ batch lower bound under sample complexity $\tilde{O}(H_I)$.

[4]We note that setting $C$ to be any positive constant will not change the asymptotic value of $R_I$. For the sake of convenience in the analysis, we choose $C = 15\sqrt{2}$.

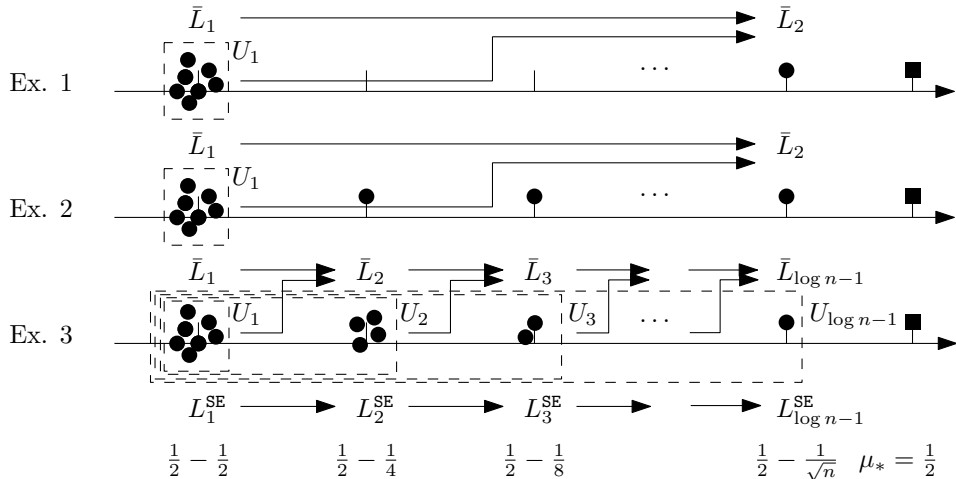

Figure 1: Visualization of the three examples. The detailed description of three examples are listed from **Ex. 1** to **Ex. 3**. Each suboptimal arm is represented by a disk (with slight shifts to avoid overlapping), while the best arm is represented by a square. $L_i^{\text{SE}}$ represents the average number of arm pulls by successive elimination, and $\bar{L}_i$ represents the average number of arm pulls by IS-SE. $U_i$ represents the set of eliminated arm in the $i$-th batch. In all three examples, successive elimination needs $\Theta(\log n)$ batches. However, for the first two examples, IS-SE only needs $O(1)$ batches. For the third example, IS-SE shares the same batch complexity as successive elimination.

$\Delta_j$). This contrasts with successive elimination, which uses fixed grids $\bar{L}_r = 4^r$; see Section 3 for a detailed explanation. It is easy to see that $\log(1/\Delta_2)$ is an upper bound of $R_I$. We will shortly give examples showing that for some instances $I$, the gap between $R_I$ and $\log(1/\Delta_2)$ can be fairly large. We further provide the following upper bound for $R_I$:

$$R_I \leq O(\alpha + \log(H_I/n)), \tag{1.3}$$

where $\alpha$ denotes the number of indices $i$ such that $U_i \neq U_{i+1}$. It is not difficult to see that both terms in (1.3) are no larger than $\log(1/\Delta_2)$.

Second, we propose a new algorithm named *Instance-Sensitive Successive Elimination (IS-SE)*, which achieves nearly optimal sample complexity $\tilde{O}(H_I)$ and batch complexity $O(R_I)$. IS-SE utilizes the arm elimination framework similar to successive elimination, but updates its batch size according to Definition 1.1. We note that due to the inherent randomness in the arm pulls, it is *not* feasible to design an algorithm that strictly follows Definition 1.1. Consequently, we cannot compute $\bar{L}_r$ values precisely, and this inaccuracy will propagate and accumulate with each successive batch. Therefore, we need new ideas to simulate the recursion process described in Definition 1.1.

Third, we extend the definition of $R_I$ to the best arm identification in linear bandits, and propose the *Instance-Sensitive RAGE (IS-RAGE)* algorithm as an counterpart of IS-SE. We show that IS-RAGE finds the best arm with nearly optimal sample complexity as well as an improved batch complexity.

**Examples.** To show that our instance-dependent batch complexity $R_I$ for MAB is always no worse than and often significantly outperform the state-of-the-art batch complexity $\log(1/\Delta_2)$, we present a few examples; see Figure 1 for their visualizations.

**Ex. 1** The best arm has mean $\frac{1}{2}$; $(n-2)$ arms have mean $\left(\frac{1}{2} - \frac{1}{2}\right)$, and 1 arm has mean $\left(\frac{1}{2} - \frac{1}{\sqrt{n}}\right)$. For this instance, $R_I = O(1)$, while $\log(1/\Delta_2) = \log\sqrt{n} = \Theta(\log n)$.

**Ex. 2** The best arm has mean $\frac{1}{2}$; $(n - \log_2 n + 1)$ arms have mean $\left(\frac{1}{2} - \frac{1}{2}\right)$; 1 arm has mean $\left(\frac{1}{2} - \frac{1}{4}\right)$; 1 arm has mean $\left(\frac{1}{2} - \frac{1}{8}\right)$; ..., and 1 arm has mean $\left(\frac{1}{2} - \frac{1}{\sqrt{n}}\right)$. For this instance, $R_I = O(1)$, while $\log(1/\Delta_2) = \log\sqrt{n} = \Theta(\log n)$.

**Ex. 3** Let $x = \frac{3n-2}{4} = \Theta(n)$. The best arm with mean $\frac{1}{2}$; $x$ arms with mean $\left(\frac{1}{2} - \frac{1}{2}\right)$; $\frac{x}{4}$ arms with mean $\left(\frac{1}{2} - \frac{1}{4}\right)$; ...; and 1 arm with mean $\left(\frac{1}{2} - \frac{1}{\sqrt{n}}\right)$. For this instance, $R_I = \tilde{O}(\log n)$, which is comparable to $\log(1/\Delta_2) = \log \sqrt{n} = \Theta(\log n)$.

**Notations and Conventions.** We use lower case letters to denote scalars and vectors, and upper case letters to denote matrices. We denote by $[n]$ the set $\{1, \ldots, n\}$. For a vector $\mathbf{x} \in \mathbb{R}^d$ and a positive semi-definite matrix $\mathbf{\Sigma} \in \mathbb{R}^{d \times d}$, we denote by $\|\mathbf{x}\|_2$ the vector's Euclidean norm and define $\|\mathbf{x}\|_{\mathbf{\Sigma}} = \sqrt{\mathbf{x}^\top \mathbf{\Sigma} \mathbf{x}}$. Unless otherwise stated, $\log x$ refers to the logarithm of $x$ with base 2.

For convenience, in MAB, given a subset of arms $A \subseteq [n]$, we define the *partial instance complexity* of $A$ to be $H(I')$, where $I' = A \cup \{\mu_*\}$.

## 2 RELATED WORK

A rich body of research exists on bandits, reinforcement learning, and online learning problems in the batched model. We review the most relevant ones to our work.

Batched best arm identification has been studied in both multi-armed bandits (Jun et al., 2016; Agarwal et al., 2017; Jin et al., 2019) and linear bandits (Fiez et al., 2019b; Soare et al., 2014a). However, the batch complexities in those algorithms are *not* instance-sensitive. Recently, Jin et al. (2023) studied settings where $\delta$ approaches $0$, deriving optimal sample and batch complexities for this setting. They also studied the finite $\delta$ scenario, noting that in this case, the batch complexity of the proposed algorithms is not instance-sensitive.

The other basic problem in bandits theory is *regret minimization*. The early work UCB2 (Auer et al., 2002) for regret minimization in multi-armed bandits can be implemented in $\log T$ batches where $T$ is the time horizon. Through a sequence of papers (Perchet et al., 2016; Gao et al., 2019; Esfandiari et al., 2019), almost optimal regret-batch tradeoffs have been established for both minimax and instance-dependent regret. Cesa-Bianchi et al. (2013) studied a setting where one can change the policy at any time. Jin et al. (2021) considered the scenario in which the time horizon $T$ is not known in advance, as well as batch algorithms in the asymptotic regret setting. Several papers (Kalkanli & Özgür, 2021; Karbasi et al., 2021; Karpov & Zhang, 2021) studied batched Thompson sampling for multi-armed bandits. Regret minimization has also been studied for linear (contextual) bandits (Esfandiari et al., 2019; Han et al., 2020; Ruan et al., 2021; Zhang et al., 2021), among which Zhang et al. (2021) obtained almost optimal regret-batch tradeoff for almost all settings.

We also note that there is a strong connection between the batched model and the *collaborative learning (CL)* model, which has gained attention in recent years. Notable works include: (Hillel et al., 2013; Tao et al., 2019) on BAI w.r.t. sample complexity and round complexity tradeoffs; Karpov et al. (2020) on top-$m$ best arm identifications; Wang et al. (2019) on regret minimization w.r.t. sample complexity and communication cost tradeoffs; and Karpov & Zhang (2023) on BAI w.r.t. sample complexity and communication cost tradeoffs; Karpov & Zhang (2024) on regret minimization w.r.t. sample complexity and round complexity tradeoffs. In particular, the batched model is essentially equivalent to the non-adaptive version of the CL model. Tao et al. (2019) showed that in the non-adaptive CL model, there is an algorithm for fixed-confidence BAI that achieves almost optimal sample complexity using $\log(1/\Delta_2)$ rounds (for worst-case input), where the round complexity is almost tight, up to a $\log\log(1/\Delta_2)$ factor. When the upper bound result is adapted to the batched model, we get an algorithm with $\log(1/\Delta_2)$ batch complexity, which is *not* instance-sensitive.

## 3 INSTANCE-DEPENDENT BATCHED ALGORITHM FOR MULTI-ARMED BANDITS

In this section, we present our batched algorithm IS-SE for MAB and analyze its complexities.

---

**Algorithm 1** Instance-Sensitive Successive Elimination (IS-SE)

---

**Require:** a set of arms $X$, confidence parameter $\beta_{\text{conf}}$, sample complexity parameter $\beta_{\text{sample}}$, and grid parameter $\beta_{\text{grid}}$.

1: Let $S_1 \leftarrow [n]$, $r \leftarrow 1$, $L_1 \leftarrow \beta_{\text{grid}}$, $\delta_1 \leftarrow \frac{3\delta}{\pi^2}$ ;
2: **while** $|S_r| > 1$ **do**
3:     **for** each arm $i \in S_r$ **do**
4:         pull arm $i$ for $L_r \log(r^2 n/\delta_1)$ times; let $\hat{p}_i^r$ be the empirical mean of arm $i$ ;
5:     **end for**
6:     let $\hat{p}_*^r \leftarrow \max_{i \in S_r} \hat{p}_i^r$, and let $\epsilon_i^r \leftarrow \hat{p}_*^r - \hat{p}_i^r$ ;
7:     delete the arm set $O_r \subseteq S_r$ such that for each $i$ in $S_r$ for which $\epsilon_i^r > \beta_{\text{conf}}/\sqrt{L_r}$ ;
8:     set $S_{r+1} \leftarrow S_r \setminus O_r$, and set

$$L_{r+1} \leftarrow \beta_{\text{grid}} L_r + \frac{\beta_{\text{sample}}}{|S_{r+1}|} \sum_{s=1}^{r} \sum_{j \in O_s} (\epsilon_j^s)^{-2} ;$$

9:     $r \leftarrow r + 1$ ;
10: **end while**
**Ensure:** the arm in $S_r$ .

---

## 3.1 THE ALGORITHM

Our algorithm is based on successive elimination (SE) with a refined scheme to update the batch sizes. We include the details of IS-SE in Algorithm 1.

We start by describing the SE algorithm and then introduce our algorithm, emphasizing the key differences. The SE algorithm can be seen as a special realization of Algorithm 1 with $\beta_{\text{sample}} = 0$. At $r$-th batch, we denote $S_r$ as the set of arms that have not been eliminated yet. In each batch, SE follows a round-robin approach, pulling each arm in the set $S_r$ for roughly $L_r$ times (up to a logarithmic factor). After the arm pulls, SE estimates the reward mean for each arm and eliminates those whose estimated mean is at least $\beta_{\text{conf}}/\sqrt{L_r}$ below the empirical best arm, where $\beta_{\text{conf}}$ is a parameter controlling the confidence level for eliminating suboptimal arms. SE then sets the number of pulls for the next batch using a simple multiplication rule: $L_{r+1} := \beta_{\text{grid}} L_r$, where $\beta_{\text{grid}} > 1$ is a parameter that controls the batch-sample complexity tradeoff.

Following the analysis in Even-Dar et al. (2002), we can show that SE finds the best arm with high probability using $\beta_{\text{grid}} \cdot \tilde{O}(H_I)$ pulls and $O(\log_{\beta_{\text{grid}}}(1/\Delta_2))$ batches. By selecting $\beta_{\text{grid}}$ as a constant, e.g., $\beta_{\text{grid}} = 4$, we recover the result in Even-Dar et al. (2002). Our main observation is that such a batch update rule may be too conservative. We can incorporate an additional term into the batch update scheme that decreases the batch complexity for many input instances without increasing the overall sample complexity.

**Our approach.** In IS-SE (Algorithm 1) at Line 8, we introduce in the $r$-th batch an additional budget $\sum_{s=1}^{r} \sum_{j \in O_s} (\epsilon_j^s)^{-2}$, which represents the estimated instance complexity of the set of eliminated arms in the first $r$ batches (i.e., arms in $[n] \setminus S_{r+1}$); we then uniformly distribute this additional budget to the set of remaining arms $S_{r+1}$, scaled by the parameter $\beta_{\text{sample}}$. Intuitively, we can think that in order to eliminate arms in $[n] \setminus S_{r+1}$, one has to spend at least $\sum_{s=1}^{r} \sum_{j \in O_s} (\epsilon_j^s)^{-2}$ pulls. Therefore, it would not be a big waste to use this amount of budget for the next batch.

## 3.2 THE ANALYSIS

The following theorem suggests that Algorithm 1 is able to find the best arm and its sample complexity only suffers an additional $\log n \min\{\log n, \log(1/\Delta_2)\}$ factor compared with the optimal sample complexity. Moreover, its batch complexity outperforms $\log(1/\Delta_2)$.

**Theorem 3.1.** *Select $\beta_{conf} = 5\sqrt{2}$, $\beta_{sample} = 25/9$ and $\beta_{grid} = 4$. With probability $1 - \delta$, Algorithm 1 satisfies the following conditions:*

*1. (Correctness) Algorithm 1 returns the best arm.*

2. *(Batch complexity) Define $\{\bar{L}_r\}$ and $\{U_r\}$ as in (1.2) (Definition 1.1). Let $R_I$ to be the minimal $r \in \mathbb{N}$ satisfying $|U_r| = n - 1$. Then the batch complexity of Algorithm 1 is bounded by $R_I$. Furthermore, suppose that $0 = r_0 \leq r_1 < \cdots < r_\alpha = R_I$ satisfying $\forall 1 \leq i \leq \alpha, U_{r_i} \neq U_{r_i+1}$. Then $R_I \leq \alpha + \log_{\frac{451}{450}}(450H_I/n) = O(\log(1/\Delta_2))$.*

3. *(Sample complexity) Algorithm 1 has a sample complexity of*

$$O\left( \left( \log(n/\delta) + \log\log \Delta_2^{-1} \right) \log\min\{n, \Delta_2^{-1}\} \cdot H_I \right).$$

We provide a proof sketch here and leave the full proof to Appendix B.

*Proof Sketch.* For sample complexity, we start by showing that with high probability, the additional term $\sum_{s=1}^{r} \sum_{j \in O_s} (\epsilon_j^s)^{-2}$ at Line 8 approximates the partial instance complexity over the eliminated arms, that is, $\sum_{s=1}^{r} \sum_{j \in O_s} \Delta_j^{-2}$, up to some constant. We then show that the additional sample budget $\sum_{s=1}^{r} \sum_{j \in O_s} (\epsilon_j^s)^{-2}$ included at each update of $L_{r+1}$ is not overly costly. Specifically, we prove that the sum of these terms only contributes an extra $\log\min\{n, \Delta_2^{-1}\}H_I$ to the overall sample complexity.

For batch complexity, we align the number of batches $r$ that IS-SE uses with the batches defined in Definition 1.1. A major challenge is to deal with the randomness introduced by the sampling steps in IS-SE. We use induction to bound $L_r$ by $\bar{L}_r$ and $\cup_{i=1}^{r} O_i$ by $U_r$.

For the inequality $R_I \leq \alpha + \log_{\frac{451}{450}}(450H_I/n)$, we show that in each batch, we either eliminate more arms (i.e., $U_r \neq U_{r+1}$), or the estimated instance complexity $\sum_{i \in [n]} (\epsilon_i^r)^{-2}$ doubles. These proof techniques may be of independent interest. $\square$

We make some remarks regarding Theorem 3.1.

**Remark 3.2.** *IS-SE finds the best arm with $\tilde{O}(H_I)$ sample complexity and at most $\alpha + \log(H_I/n)$ batch complexity. Note that $\alpha \leq \log(1/\Delta_2)$ due to the fact that $L_{r+1} > 4L_r$ and $H_I/n \leq 1/\Delta_2^2$. Therefore, the batch complexity is always no more than $\log(1/\Delta_2)$.*

**Remark 3.3.** *Although IS-SE achieves both instance-sensitive sample complexity $\tilde{O}(H_I)$ and batch complexity $R_I$, it does not need either $H_I$ or $R_I$ as its input.*

**Remark 3.4.** *One might argue that IS-SE is not necessarily superior to SE because its sample complexity exceeds $H_I$ by some logarithmic factors, and similar improvements in batch complexity might be achieved by adjusting the $\beta_{grid}$ parameter in SE. However, for any choice of $\beta_{grid} = \text{poly}\log(n/(\delta\Delta_2))$, the batch complexity of SE can only be bounded by $O(\log_{\beta_{grid}}(1/\Delta_2)) = O(\log(1/\Delta_2)/\log\log(n/(\delta\Delta_2)))$, which for many input instances, such as **Ex. 1** and **Ex. 2**, is still much worse than $R_I$.*

## 4 INSTANCE-DEPENDENT BATCH COMPLEXITY FOR LINEAR BANDITS

In this section, we extend our instance-dependent batched MAB algorithm to the linear bandits setting. In best arm identification in linear bandits (BAI-L), we again have a set of $n$ arms, but now each arm is associated with a $d$-dimensional attribute vector. Let $x_i \in \mathbb{R}^d$ denote the context associated with the $i$-th arm, and we use $X$ to denote the collection of all contexts. We assume $\|x_i\|_2 \leq 1$ for all arms. Given an error probability $\delta$, our goal is to identify an arm $x_* = \arg\max_{x \in X} x^\top \theta^*$ with probability $(1 - \delta)$ using the smallest number of arm pulls, where $\theta^* \in \mathbb{R}^d$ is an unknown vector representing the hidden linear model, and each pull of an arm $x$ returns a value $x^\top \theta^* + \epsilon$, where $\epsilon \sim \mathcal{N}(0, 1)$.

We introduce a few more notations. Let $\mathcal{Y}(S) := \{x - x' : \forall x, x' \in S, x \neq x'\}$ and let $\mathcal{Y}^*(S) := \{x_* - x : \forall x \in S \setminus \{x_*\}\}$. We will use the following optimal design for a given set $A$ and its corresponding value.

**Definition 4.1.** *Given the arm set $X \subseteq \mathbb{R}^d$ satisfying $|X| = n$ and a test set $A \subseteq \mathbb{R}^d$, we define the optimal design of $A$ by $\lambda(A)$, which is*

$$\lambda(A) := \underset{\lambda \in \Delta^X}{\arg\min} \max_{y \in A} \|y\|^2_{(\sum_{x \in X} \lambda_x x x^\top)^{-1}}, \quad where \quad \Delta^X := \left\{\lambda \in \mathbb{R}^{|X|} : \lambda \geq 0, \sum_{x \in X} \lambda_x = 1\right\}.$$

---

**Algorithm 2** Instance-Sensitive RAGE (IS-RAGE)

---

**Require:** a set of arms $X$, confidence parameter $\beta_{\text{conf}}$, sample complexity parameter $\beta_{\text{sample}}$, and grid parameter $\beta_{\text{grid}}$.

1: Let $r = 1$, $S_1 = X$, $L_1 = \beta_{\text{grid}}$, $\lambda_1^* \leftarrow \lambda(\mathcal{Y}(S_1))$, $\rho_1 \leftarrow \rho(\mathcal{Y}(S_1))$ ;
2: **while** $|S_r| > 1$ **do**
3:     set $\delta_r \leftarrow \delta/r^2$ and $N_r \leftarrow 4 \max\{2\log(|S_r|^2/\delta_r)\rho_r L_r, d\}$ ;
4:     set $\{x_r^1, \ldots, x_r^{N_r}\} \leftarrow \text{Round}(\lambda_r^*, N_r, X, \mathcal{Y}(S_r))$, where Round() is defined in Definition 4.2 ;
5:     pull $x_r^1, \ldots, x_r^{N_r}$ and obtain observed rewards $c_r^i \sim \mathcal{N}((\theta^*)^\top x_r^i, 1)$; let

$$\theta_r \leftarrow \left( \sum_{i=1}^{N_r} x_r^i (x_r^i)^\top \right)^{-1} \sum_{i=1}^{N_r} x_r^i c_r^i \; ;$$

6:     set $\hat{p}_x^r \leftarrow \theta_r^\top x$, $\hat{p}_*^r \leftarrow \max_{x \in S_r} \hat{p}_x^r$, and $\epsilon_x^r \leftarrow \hat{p}_*^r - \hat{p}_x^r$ ;
7:     set the elimination set $O_r \leftarrow \{x \in S_r | \epsilon_x^r \geq \beta_{\text{conf}}/\sqrt{L_r}\}$, and set $\epsilon_x \leftarrow \epsilon_x^r$ ;
8:     let $S_{r+1} \leftarrow S_r \setminus O_r$, $\lambda_{r+1}^* \leftarrow \lambda(\mathcal{Y}(S_{r+1}))$, and $\rho_{r+1} \leftarrow \rho(\mathcal{Y}(S_{r+1}))$ ;
9:     let $T_r$ satisfy that $\beta_{\text{grid}}^{T_r} \leq L_r < \beta_{\text{grid}}^{T_r+1}$ and set

$$L_{r+1} \leftarrow \beta_{\text{grid}} L_r + \frac{\sum_{t=1}^{T_r} \beta_{\text{grid}}^t \cdot \overbrace{\rho(\mathcal{Y}(X \setminus \{x \in \cup_{s=1}^r O_s : \epsilon_x > \beta_{\text{sample}} \cdot \sqrt{\beta_{\text{grid}}^{-t}}\}))}^{I_t}}{\rho(\mathcal{Y}(X \setminus \cup_{s=1}^r O_s))} \; ; \quad (4.1)$$

10:     $r \leftarrow r + 1$ ;
11: **end while**
**Ensure:** the arm in $S_r$ .

---

*We also define the* optimal design value *of $A$ by* $\rho(A) := \max_{y \in A, \lambda = \lambda(A)} \|y\|^2_{(\sum_{x \in X} \lambda_x x x^\top)^{-1}}$ .

Our algorithm relies on the *Round* function introduced by (Allen-Zhu et al., 2021; Fiez et al., 2019a). This function outputs a multiset of size $N$ from the input arm set $X$ with specific approximation guarantees.

**Definition 4.2** (Allen-Zhu et al. (2021); Fiez et al. (2019a) ). *There is a function Round$(\lambda, N, X, Y)$ that outputs a multiset $(x_1, ..., x_N)$, where $x_i \in X$ for all $i \in [N]$ and $N$ is the total number of arms, and $Y$ is the measurement set. If $N > d$, then the output multiset satisfies*

$$\max_{y \in Y} \|y\|^2_{(\sum_i^N x_i x_i^\top)^{-1}} \leq \frac{2}{N} \max_{y \in Y} \|y\|^2_{(\sum_{x \in X} \lambda_x x x^\top)^{-1}}.$$

We define a value $\psi^*$ to represent the instance complexity of BAI-L, analogous to how $H_I$ characterizes the instance complexity for BAI-M.

**Definition 4.3** (Soare et al. (2014b); Fiez et al. (2019a)). *We define $\psi^* \triangleq \psi^*(X)$ as follows:*

$$\psi^* := \min_{\lambda \in \Delta^X} \max_{x \in X \setminus \{x_*\}} \frac{\|x - x_*\|^2_{(\sum_{x' \in X} \lambda_{x'} x'(x')^\top)^{-1}}}{(x_* - x)^\top \theta^*}.$$

It has been shown that $\psi^*$ is the *instance-sensitive* lower bound of the instance complexity of BAI-L, with certain existing algorithms like RAGE achieving this bound up to logarithmic factors (Soare et al., 2014b; Fiez et al., 2019a).

## 4.1 THE ALGORITHM

Our algorithm IS-RAGE is described in Algorithm 2, which is built upon the RAGE algorithm proposed by Fiez et al. (2019a).

RAGE can essentially be seen as an implementation of Algorithm 2 with $\beta_{\text{sample}} = 0$. At round $r$, RAGE maintains a set $S_r$ that includes the optimal arm $x_*$ with high probability. It allocates a budget

$N_r$ based on observations from the previous $(r-1)$ rounds, and then calls the Round function (as defined in Definition 4.2) to generate a multiset of arms $X_r = \{x_r^1, \ldots, x_r^{N_r}\}$. RAGE pulls each arm in $X_r$ and computes $\theta_r$ using ordinary least-squares (OLS) over the arm set and the reward set. Based on $\theta_r$, it calculates the suboptimality gap $\epsilon_x^r$ for each $x \in S_r$ and eliminates arms with large $\epsilon_x^r$. RAGE then continues to the next round with a reduced arm set $S_{r+1} = S_r \setminus O_r$, where $O_r$ is the set of arms eliminated in round $r$.

**Our approach.** Similar to IS-SE, IS-RAGE (Algorithm 2) introduced an additional sample budget term at (4.1) when allocating the sample budget per arm for the next batch. The first term in the right hand side of (4.1) is the same as that in RAGE, which aims to double the current number of pulls for each arm. For the second term, we uniformly distribute the *estimated* partial instance complexity of all eliminated arms (represented by $I_t$ in (4.1)) to the set of remaining arms $S_{r+1}$, where $\rho(\mathcal{Y}(X \setminus \cup_{s=1}^r O_s))$ represents the "effective" cardinality of $S_{r+1}$ for BAI-L. To better understand why $I_t$ approximates the partial instance complexity of all eliminated arms, we have the following lemma.

**Lemma 4.4.** *For any $L > 0$, we have $L \cdot \rho(\mathcal{Y}(z \in X : \Delta_z \leq 15/\sqrt{L})) \leq 900\psi^*$, where $\psi^*$ is defined in Definition 4.3.*

Roughly speaking, by selecting $L = \beta_{\text{grid}}^t$, Lemma 4.4 shows that $I_t$ approximates the total instance complexity $\psi^*$ over the whole arm set $X$, which naturally serves as an upper bound of the partial instance complexity of eliminated arms $x \in \cup_{s=1}^r O_s$. We claim that such an algorithm design is necessary for linear bandits due to the fact that arms in linear bandits are often *linearly dependent*, while arms in MAB are *independent*.

**Remark 4.5.** *We note that the hyperparameter $\beta_{sample}$ governs how the updated $L_{r+1}$ relates to the partial instance complexity of the eliminated arms. As $\beta_{sample}$ increases, each $I_t$ in (4.1) also scales up, functioning similarly to $\beta_{sample}$ in IS-SE.*

## 4.2 THE ANALYSIS

We try to show that IS-RAGE achieves the instance complexity $\psi^*$ up to some logarithmic factors, with a better batch complexity. Similar to Definition 1.1, we define the instance-sensitive batch complexity for linear bandits as follows. For simplicity, we abuse the notations and reuse $R_I$, $L_r$, $\bar{L}_r$ for linear bandits.

**Definition 4.6.** *Define a sequence $\bar{L}_r = 4^{\bar{T}_r}$ for some integer $\bar{T}_r$ as follows. Let $U_r := \{x|\Delta_x > 15/\sqrt{\bar{L}_r}\}$ and*

$$\hat{L}_{r+1} \leftarrow 4\bar{L}_r + \frac{\sum_{t=1}^{\bar{T}_r} 4^t \cdot \rho(\mathcal{Y}(X \setminus \{x \in U_r : \Delta_x > 15 \cdot 2^{-t}\}))}{\rho(\mathcal{Y}(X \setminus U_r))}, \tag{4.2}$$

*and we select $\bar{L}_{r+1} = 4^T$ satisfying $4^T \leq \hat{L}_{r+1} < 4^{T+1}$. Denote $R_I$ to be the minimal $r$ satisfying $U_r = X$.*

Generally speaking, the second term of the RHS in (4.2) gives an approximation of the partial instance complexity of arms in $U_r$. Ideally, we would like to use a definition similar to $\psi^*$ to directly define the partial instance complexity of arms in $U_r$. However, since $x_*$ is unknown to the agent, we are unable to compute $\psi^*$ directly.

We have the following theorem regarding Algorithm 2,

**Theorem 4.7.** *With probability at least $1 - \delta$, Algorithm 2 satisfies the following conditions:*

*1. (Correctness) Algorithm 2 returns the optimal arm $\mu_1$.*

*2. (Batch complexity) Algorithm 2 runs at most $R_I$ batches where $R_I$ is defined in Definition 4.6. Furthermore, $R_I$ can be bounded as*

$$R_I \leq \alpha + \log_{\frac{5}{4}} \frac{900 \log_2(1/\Delta_2)\psi^*}{\rho(\mathcal{Y}^*(X))},$$

*where $\alpha$ is the number of different $U_r$'s, and $\psi^*$ is defined in Definition 4.3.*

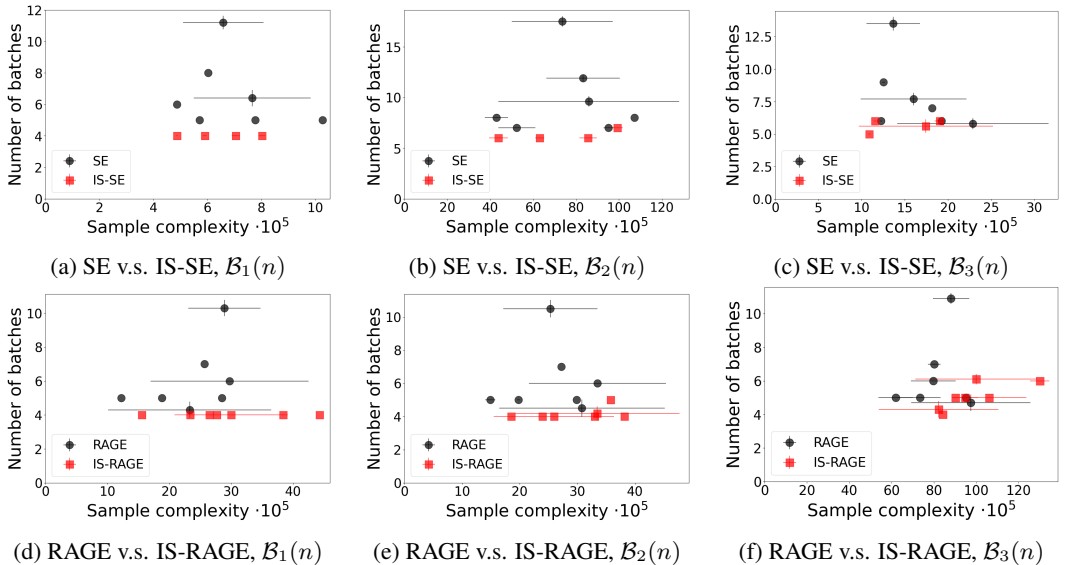

(a) SE v.s. IS-SE, $\mathcal{B}_1(n)$     (b) SE v.s. IS-SE, $\mathcal{B}_2(n)$     (c) SE v.s. IS-SE, $\mathcal{B}_3(n)$

(d) RAGE v.s. IS-RAGE, $\mathcal{B}_1(n)$     (e) RAGE v.s. IS-RAGE, $\mathcal{B}_2(n)$     (f) RAGE v.s. IS-RAGE, $\mathcal{B}_3(n)$

Figure 2: Sample complexity v.s. number of batches.

*3. (Sample complexity) Algorithm 2 has a sample complexity of*

$$O(\psi^* \log^2(1/\Delta_2) \log(|X|/\delta) + \log(1/\Delta_2)d).$$

We provide a proof sketch here and leave the full proof to Appendix C. The proof follows a similar approach to that of Theorem 3.1, but involves more technical details due to the added complexity of linear bandits.

*Proof Sketch.* Similar to the proof to IS-SE, we first prove that $\epsilon_x^r$ serves as a good approximation of the suboptimality gap $\Delta_x$ under a high probability event. For batch complexity, we align the number of batches $r$ with the batches $R_I$ we have defined in Definition 4.6. We still use induction to bound $L_r \geq \bar{L}_r$ and $U_r \subseteq \cup_{s=1}^r O_s$ to deal with the randomness introduced by the sampling steps in IS-RAGE. Unlike IS-SE, due to the definition of $I_t$, we force $\bar{L}_r$ to be the power of 4 to make sure the induction step can be proceeded, which makes our analysis slightly different from that of IS-SE. To show that $R_I \leq \alpha + \log_{\frac{5}{4}} \frac{900 \log_2(1/\Delta_2)\psi^*}{\rho(\mathcal{Y}^*(X))}$, we adapt a similar strategy as IS-SE with a new bound on the partial instance complexity of linear bandits. Finally, to prove the sample complexity result, we show that the additional sample complexity introduced by the summation of $I_t$ can be upper bounded by $\text{polylog}(|X|/\delta)\psi^*$, which suggests that the sample complexity of IS-RAGE is the same as $\psi^*$ up to some logarithmic factors. □

**Remark 4.8.** *Theorem 4.7 shows that IS-RAGE can find the best arm with $\tilde{O}(\psi^*)$ sample complexity, which is only $\log(1/\Delta_2)$ worse than RAGE Fiez et al. (2019a). Meanwhile, it only requires $O(\alpha + \log \frac{\psi^*}{\rho(\mathcal{Y}^*(X))} + \log\log(1/\Delta_2))$ number of batches. We can verify that both $\alpha$ and $\log \frac{\psi^*}{\rho(\mathcal{Y}^*(X))}$ are smaller than $\log(1/\Delta_2)$, which suggests that our bound on the batch complexity is tighter.*

## 5 EXPERIMENTS

We run experiment on both synthetic data and real-world data [5] to validate the effectiveness of our proposed algorithm.

In the following, we test our proposed algorithms (IS-SE and IS-RAGE) with baseline algorithms (SE and RAGE) over synthetic data here.

---

[5]Due to limited space, please refer to Appendix A for the experimental results on real-world data.

**Bandit setting.** We evaluate our algorithm using the three examples introduced earlier. These examples are denoted as $\mathcal{B}_1(n)$, $\mathcal{B}_2(n)$, and $\mathcal{B}_3(n)$, corresponding to **Ex. 1**, **Ex. 2**, and **Ex. 3**, respectively, where $n$ represents the number of arms in each case. When the agent selects an arm with a mean reward $\alpha$, the environment returns an observation $\alpha + \epsilon$, where $\epsilon \sim N(0, 0.1)$. For the multi-armed bandit setting, we test our algorithm on each $\mathcal{B}_i(n)$ with $n = 1000$. In the linear bandit setting, we set the dimension $d = n$ and define the arm set as $X = \{e_i\}_{i=1}^n$, where $e_i$ represents the $i$-th basis vector in $n$-dimensional space. Due to memory constraints, we test the linear bandit algorithms with $n = 500$.

**Algorithms.** We compare IS-SE (Algorithm 1) with SE across different parameter configurations on various bandit instances. For both SE and IS-SE, we set the confidence level $\delta = 0.1$. Specifically, we evaluate SE with different values of $\beta_{\text{grid}}$, and IS-SE with varying selections of both $\beta_{\text{grid}}$ and $\beta_{\text{sample}}$. Note that SE is equivalent to IS-SE when $\beta_{\text{sample}} = 0$. For SE, we fix $\beta_{\text{conf}} = 1$ and test it with $\beta_{\text{grid}} \in \{2, 3, ..., 8\}$. For IS-SE, we also set $\beta_{\text{conf}} = 1$ and test it with $\beta_{\text{grid}} \in \{2, 3, ..., 8\}$ and $\beta_{\text{sample}} \in \{0.5, 1, 1.5, 2\}$.

We also compare IS-RAGE (Algorithm 2) with RAGE. Specifically, for both RAGE and IS-RAGE, we set the $\beta_{\text{conf}} = 1$ and the confidence level $\delta = 0.1$. Note that RAGE is equivalent to IS-RAGE when $\beta_{\text{sample}} = 0$. For RAGE, we test it with $\beta_{\text{grid}} \in \{2, 3, ..., 8\}$. For IS-RAGE, we test it with $\beta_{\text{grid}} \in \{2, 3, ..., 8\}$ and $\beta_{\text{sample}} \in \{0.5, 1, 1.5, 2\}$.

For each parameter set, we run the algorithms 10 times, reporting the mean and variance of the number of batches and sample complexity needed to identify the best arm.

**Results.** The results are recorded in Figure 2. For simplicity, we report the results of IS-SE with $\beta_{\text{grid}} = 5$ for all three instances, and we report the results of IS-RAGE with $\beta_{\text{sample}} = 1$ as well. In the first two bandit instances $\mathcal{B}_1(n)$ and $\mathcal{B}_2(n)$, our IS-SE and IS-RAGE consistently outperforms SE and RAGE. This indicates that, for the same sample complexity, IS-SE and IS-RAGE are more efficient on allocating samples across batches, resulting in lower batch complexity compared to SE and RAGE. This aligns with our analysis in the introduction, where we showed that IS-SE achieves an $O(1)$ batch complexity, while SE has an $O(\log n)$ batch complexity. For the third bandit instance $\mathcal{B}_3(n)$, the results of SE and IS-SE can hardly be separated, so do RAGE and IS-RAGE. This suggests IS-SE and IS-RAGE do not yield a better sample-batch tradeoff compared with SE and RAGE, which also aligns our analysis in the introduction.

## 6 CONCLUSION AND FUTURE WORK

In this paper, we have proposed algorithms for best arm identification in multi-armed and linear bandits, achieving near-optimal sample complexities along with instance-sensitive batch complexities. Our batch complexities can be significantly better than $\log(1/\Delta_2)$ for many input instances, where the latter is minimax-optimal up to a double-logarithmic factor. Given the importance of batched algorithms in online learning, we believe that exploring algorithms with instance-sensitive batch complexity could be an interesting direction for other bandit and online learning problems. A couple of immediate questions arise from this work. First, it would be valuable to analyze the *average* batch complexity of our algorithms on typical input distributions. Second, it would be beneficial to run our experiments on larger datasets to better assess their practicality in real-world applications.

## ACKNOWLEDGMENT

Qin Zhang was supported in part by NSF CCF-1844234. Tianyuan Jin was supported by the Singapore Ministry of Education AcRF Tier 2 grant (A-8000423-00-00).

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

## A    EXPERIMENTAL RESULTS ON REAL DATA

In this section, we provide the experimental results on the real-world data set.

We evaluate our algorithms on the Movielens25M dataset[6], which contains 25 million ratings for 60,000 movies by 160,000 users. We formulate this as a multi-armed bandit (MAB) problem by selecting the top 1,000 movies with the most ratings and treating each movie as an arm. The reward for pulling an arm corresponds to the negative rating given by a user for the associated movie. To accelerate the sampling process, we consider only the first 50 ratings for each movie across all users.

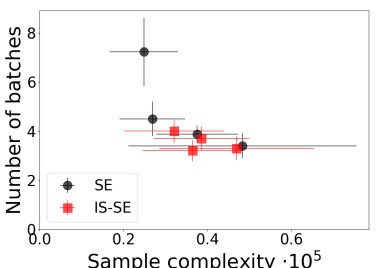

Figure 3: SE v.s. IS-SE on Movielens25M

We compare SE and IS-SE on the Movielens25M dataset. Each algorithm is tested with $\beta_{\text{grid}} \in \{2, 4, 6, 8\}$ and $\beta_{\text{sample}} \in \{0.5, 1, 1.5, 2\}$. For each parameter configuration, we run the algorithms 10 times and report the mean and variance of the number of batches and the sample complexity required to identify the best arm. The results are shown in Figure 3. We observe that IS-SE consistently outperforms SE in terms of the sample complexity-batch complexity trade-off, highlighting the effectiveness of our algorithm.

## B    PROOF OF THEOREM 3.1

We now prove Theorem 3.1. We need the following technical lemma which characterizes the concentration properties of subgaussian random variables.

**Lemma B.1** (Corollary 5.5 in Lattimore & Szepesvári (2020)). *Assume that $X_1, \ldots, X_n$ are independent, $\sigma$-subguassian random variables centered around $\mu$. Then for any $\epsilon > 0$*

$$\mathbb{P}(\hat{\mu} \geq \mu + \epsilon) \leq \exp\left(-\frac{n\epsilon^2}{2\sigma^2}\right) \quad and \quad \mathbb{P}(\hat{\mu} \leq \mu - \epsilon) \leq \exp\left(-\frac{n\epsilon^2}{2\sigma^2}\right),$$

*where $\hat{\mu} = 1/n \sum_{t=1}^{n} X_t$. We often use the following equalivent inequality: we have*

$$\mathbb{P}\left(|\hat{\mu} - \mu| \leq \sqrt{\frac{2\sigma \log(2/\delta)}{n}}\right) \geq 1 - \delta.$$

Next we start our main proof. We divide our proof into two parts. In the first part, we prove that our algorithm can find the optimal arm, and we bound the sample complexity for our algorithm to find the optimal arm. In the second part, we prove that our algorithm enjoys a better batch complexity compared with the vanilla $\log(1/\Delta_2)$.

### B.1    PROOF OF CORRECTNESS OF THEOREM 3.1

*Proof of Correctness of Theorem 3.1.* For any fixed arm $i$ at batch $r$, we have

$$\mathbb{P}\left(|\hat{p}_i^r - \mu_i| \geq \sqrt{\frac{2}{L_r}}\right) \leq 2\exp\left(-\frac{1}{2} \cdot L_r \cdot \frac{2\log(r^2 n/\delta_1)}{L_r}\right) \leq \frac{2\delta_1}{r^2 n}.$$

---

[6]`https://grouplens.org/datasets/movielens/25m/`

Applying the union bound over all arms and all batches, we have

$$\mathbb{P}\left(\bigcap_{r\geq 1}\bigcap_{i\in[n]}\left\{|\hat{p}_i^r - \mu_i| \geq \sqrt{\frac{2}{L_r}}\right\}\right) \leq \sum_{r>1}\left(n\cdot\frac{2\delta_1}{r^2 n}\right) = \frac{\pi^2\delta_1}{3}.$$

Let $\mu_{*_r} = \arg\max_{i\in S_r}\hat{p}_i^r$. Therefore, with probability at least $\pi^2\delta_1/3$, we have for all $r > 1$,

$$\hat{p}_*^r - \sqrt{\frac{2}{L_r}} \leq \mu_{*_r} \leq \mu_1 \leq \hat{p}_1^r + \sqrt{\frac{2}{L_r}},$$

Therefore, with probability at least $1 - \pi^2\delta_1/3 = 1 - \delta$, Algorithm 1 will return the best arm.

For simplicity, we denote the event defined in (B.1) as $\mathcal{E}$, i.e.,

$$\mathcal{E} := \bigcap_{r\geq 1}\bigcap_{i\in[n]}\left\{|\hat{p}_i^r - \mu_i| \geq \sqrt{\frac{2}{L_r}}\right\}.$$

$\square$

## B.2 PROOF OF BATCH COMPLEXITY OF THEOREM 3.1

*Proof of Batch Complexity of Theorem 3.1.* We condition on event $\mathcal{E}$. First, by Line 8 of Algorithm 1 and the fact that $\frac{1}{3}\Delta_j \leq \epsilon_j^r \leq \frac{5}{3}\Delta_j$, we have

$$L_{r+1} \geq 4L_r + f(\cup_{s=1}^r O_s) \text{ where } f(A) := \frac{\sum_{j\in A}1/\Delta_j^2}{n - |A|}.$$

Recall the two "virtual" sequences $\{\bar{L}_r\}$ and $\{U_r\}$ we have defined in (1.2). We prove by induction that for all $r$,

$$U_r \subseteq \cup_{s=1}^r O_s \quad \text{and} \tag{B.1}$$
$$L_r \geq \bar{L}_r. \tag{B.2}$$

Assuming that both (B.1) and (B.2) hold for step $r$, we first prove that (B.2) holds for step $(r+1)$:

$$L_{r+1} \geq 4L_r + f(\cup_{s=1}^r O_s) \geq 4\bar{L}_r + f(U_r) = \bar{L}_{r+1},$$

where the second inequality uses the induction hypothesis.

We next prove that (B.1) holds for step $(r+1)$. To this end, we consider the set of eliminated arms $O_{r+1}$. Note that by our algorithm design, $O_{r+1}$ satisfies

$$O_{r+1} := \{j \in [n]\setminus\cup_{s=1}^r O_s \mid \epsilon_j^{r+1} > 5\sqrt{2}/\sqrt{L_{r+1}}\}.$$

We also know that for any $j \in U_{r+1}$, $j$ satisfies that $\Delta_j > 15\sqrt{2/\bar{L}_{r+1}}$ based on the definition of $U_{r+1}$. By (B.2), we have $\Delta_j > 15\sqrt{2/L_{r+1}}$. By $\epsilon_j^{r+1} \geq \frac{1}{3}\Delta_j$, we further have $\epsilon_j^{r+1} > 5\sqrt{2/L_{r+1}}$.

We now consider any $j \in U_{r+1}$. If $j \in \cup_{s=1}^r O_s$, then we directly have $j \in \cup_{s=1}^{r+1}O_s$. Otherwise, we have

$$j \in ([n]\setminus\cup_{s=1}^r O_s)\cap\{j : \epsilon_j^{r+1} > 5\sqrt{2/L_{r+1}}\} = O_{r+1},$$

which again suggests $j \in \cup_{s=1}^{r+1}O_s$. Therefore, (B.1) holds for $(r+1)$. Consequently, both (B.1) and (B.2) hold for every $r > 0$.

By (B.2), the total number of batches can be bounded by the smallest $r$ satisfying $\bar{L}_r \geq 1/\Delta_2^2$. Note that $\bar{L}_r$ is a fixed sequence. Let $R$ be the smallest $r$ satisfying $\bar{L}_r \geq 1/\Delta_2^2$, which is the batch complexity we want to further bound.

Consider the quantity

$$H_r := \bar{L}_r(n - |U_r|) + \sum_{j\in U_r}\frac{450}{\Delta_j^2}, \text{where } U_r := \{j : \Delta_j > 15\sqrt{2/\bar{L}_r}\}.$$

By definition of $H_r$, we have $H_0 = n$ and $H_{R+1} = 450H$.

We first observe that $\{H_r\}$ is an increasing sequence.

$$\bar{L}_{r+1}(n - |U_{r+1}|) + \sum_{j \in U_{r+1}} \frac{450}{\Delta_j^2}$$

$$\geq \quad \bar{L}_r(n - |U_{r+1}|) + \sum_{j \in U_{r+1}} \frac{450}{\Delta_j^2}$$

$$= \quad \bar{L}_r(n - |U_r|) + \bar{L}_r(|U_r| - |U_{r+1}|) + \sum_{j \in U_r} \frac{450}{\Delta_j^2} + \sum_{j \in U_{r+1}/U_r} \frac{450}{\Delta_j^2}$$

$$\geq \quad \bar{L}_r(|U_r| - |U_{r+1}|) + (|U_r| - |U_{r+1}|)\bar{L}_r + \bar{L}_r(n - |U_r|) + \sum_{j \in U_r} \frac{450}{\Delta_j^2}$$

$$\geq \quad \bar{L}_r(n - |U_r|) + \sum_{j \in U_r} \frac{450}{\Delta_j^2},$$

where the first inequality holds since $\{\bar{L}_r\}$ is an increasing sequence, and the second inequality holds since for any $j \notin U_r$, we have $\Delta_j \leq 15\sqrt{2/\bar{L}_r}$ by the definition of $U_r$.

We next show that when $U_{r+1} = U_r$, we have that $H_{r+1} \geq \frac{451}{450} H_r$. Note that

$$\bar{L}_{r+1}(n - |U_{r+1}|) + \sum_{j \in U_{r+1}} \frac{450}{\Delta_j^2}$$

$$= \quad \bar{L}_{r+1}(n - |U_r|) + \bar{L}_{r+1}(|U_r| - |U_{r+1}|) + \sum_{j \in U_r} \frac{450}{\Delta_j^2} + \sum_{j \in U_{r+1}/U_r} \frac{450}{\Delta_j^2}$$

$$= \quad 4\bar{L}_r(n - |U_r|) + \sum_{j \in U_r} \frac{451}{\Delta_j^2} + \bar{L}_{r+1}(|U_r| - |U_{r+1}|) + \sum_{j \in U_{r+1}/U_r} \frac{450}{\Delta_j^2}$$

$$= \quad 4\bar{L}_r(n - |U_r|) + \sum_{j \in U_r} \frac{451}{\Delta_j^2}$$

$$\geq \quad \frac{451}{450}\left(\bar{L}_r(n - |U_r|) + \sum_{j \in U_r} \frac{450}{\Delta_j^2}\right).$$

Therefore, suppose that $0 = r_0 \leq r_1 < \cdots < r_\alpha = R$ satisfying $\forall 1 \leq i \leq \alpha, U_{r_i} \neq U_{r_i+1}$. For any $1 \leq i \leq \alpha$ and any $r_i \leq r < r_{i+1}$, we have

$$H_{r_{i+1}} \geq \frac{451}{450} H_{r_{i+1}-1} \geq \cdots \geq \left(\frac{451}{450}\right)^{r_{i+1}-r_i-1} H_{r_i+1} \geq \left(\frac{451}{450}\right)^{r_{i+1}-r_i-1} H_{r_i},$$

which gives

$$r_{i+1} - r_i \leq 1 + \log H_{r_{i+1}} - \log H_{r_i}.$$

Taking the summation, we have $R \leq \alpha + \log_{\frac{451}{450}}(450H/n)$. $\qquad \square$

### B.3 PROOF OF SAMPLE COMPLEXITY OF THEOREM 3.1

*Proof of Sample Complexity of Theorem 3.1.* We derive the sample complexity under event $\mathcal{E}$. We know that at the $r$-th batch, all $j \in S_r$ satisfy

$$|\epsilon_j^r - \Delta_j| \leq |\hat{p}_*^r - \mu_*| + |\hat{p}_j^r - \mu_j| \leq 2\sqrt{\frac{2}{L_r}}.$$

Thus, for all $j \in O_r$, we always have

$$\Delta_j \geq \epsilon_j^r - |\epsilon_j^r - \Delta_j| \geq 3\sqrt{\frac{2}{L_r}},$$

which means $\frac{1}{3}\Delta_j \le \epsilon_j^r \le \frac{5}{3}\Delta_j$. Note that $L_{r+1} \ge 4L_r$ guarantees that the number of arm pulls in $S_{r+1}$ at the $(r+1)$-th batch is at least 4 times larger than that at the $r$-th batch. Assume arm $i$ is eliminated at the $r_i$-th batch. The total number of pulls of arm $i$ can be bounded by

$$N_i \le 2\log(r_i^2 n/\delta_1)L_{r_i} \le O\big((\log(n/\delta) + \log\log\Delta_2^{-1})L_{r_i}\big).$$

Therefore, we only need to bound $\sum_{i=1}^n L_{r_i}$.

We divide $L_{r_i}$ into two parts $I_1^i$ and $I_2^i$, and bound them separately.

$$L_{r_i} \le \underbrace{\sum_{s=1}^{r_i-1} \sum_{j\in O_s} \frac{1}{|S_{r_i}|[\epsilon_j^s]^2}}_{I_1^i} + \underbrace{4L_{r_i-1}}_{I_2^i}.$$

*Bounding term $\sum_{i>1} I_1^i$.* We take the summation of $I_1^i$ $(i = 2, \ldots, n)$ to bound the total sample complexity:

$$\sum_{i=2}^n I_1^i = \sum_{i=2}^n \sum_{s=1}^{r_i-1} \sum_{j\in O_s} \frac{1}{|S_{r_i}|[\epsilon_j^s]^2} \le 9 \sum_{i=2}^n \sum_{s=1}^{r_i-1} \sum_{j\in O_s} \frac{1}{|S_{r_i}|\Delta_j^2},$$

where the inequality holds since for all $j \in O_s$, we have $\frac{1}{3}\Delta_j \le \epsilon_j^s \le \frac{5}{3}\Delta_j$. Next, note that $|S_r| = |O_r| + \ldots + |O_B|$, where $B$ is the total number of batches. We have

$$\sum_{i=2}^n \sum_{s=1}^{r_i-1} \sum_{j\in O_s} \frac{1}{|S_{r_i}|\Delta_j^2} = \sum_{s=1}^{B-1} \frac{|O_s|}{|S_s|} \sum_{t=1}^s \sum_{j\in O_t} \frac{1}{\Delta_j^2} \le \sum_{s=1}^{B-1} \frac{|O_s|}{|S_s|} \sum_{j>1} \frac{1}{\Delta_j^2} \le \min\{\log n, B\} \sum_{j>1} \frac{1}{\Delta_j^2},$$

where the last inequality holds since for each $s$, $|O_s| \le |S_s|$, and

$$\frac{|O_s|}{|S_s|} \le \sum_{k=0}^{|O_s|-1} \frac{1}{|S_s| - k}.$$

*Bounding term $\sum_{i>1} I_2^i$.* By the definition of $r_i$ and under the event $\mathcal{E}$, we have

$$\Delta_i \le \epsilon_i^{r_i-1} + |\epsilon_i^{r_i-1} - \Delta_i| \le 5\sqrt{\frac{2}{L_{r_i-1}}} + 2\sqrt{\frac{2}{L_{r_i-1}}} = 7\sqrt{\frac{2}{L_{r_i-1}}},$$

where the second inequality holds because the $i$-th arm will not be eliminated at the $(r_i - 1)$-th batch. Therefore, we have $L_{r_i-1} \le \frac{2}{\Delta_i^2}$, which suggests

$$\sum_{i>1} I_2^i \le O\bigg(\sum_{i>1} \frac{1}{\Delta_i^2}\bigg).$$

Combining the two parts and using the fact that $r_i = O(\log(1/\Delta_2))$, the total sample complexity is bounded by

$$
\begin{aligned}
2\sum_{i\ge 2} L_{r_i} \cdot \log(r_i^2 n/\delta_1) &\le O\bigg(\big(\log(n/\delta) + \log\log\Delta_2^{-1}\big)\sum_{i\ge 2} L_{r_i}\bigg) \\
&\le O\bigg(\big(\log(n/\delta) + \log\log\Delta_2^{-1}\big)\log\min\{n, \Delta_2^{-1}\}\sum_{i\ge 2} \frac{1}{\Delta_2^2}\bigg).
\end{aligned}
$$

$\square$

## C  PROOF OF THEOREM 4.7

Like Theorem 3.1, we prove our theorem for its correctness, batch complexity and sample complexity. We have the following lemma:

**Lemma C.1** (Section 2.2 in Fiez et al. (2019a)). *Given a set $Y \subseteq \mathbb{R}^d$, for any fixed design $x_1, ..., x_N$ (multiset) as well as their stochastic reward $R_1, ..., R_N$ satisfying $R_i \sim \theta_*^\top x_i + \mathcal{N}(0,1)$, let $\hat{\theta}_t$ be the OLS estimate. Then*

$$\mathbb{P}\left( \forall y \in Y, y^\top(\theta_* - \hat{\theta}_t) \geq \|y\|_{(\sum_{i=1}^N x_i x_i^\top)^{-1}} \cdot 2\sqrt{\log(|Y|/\delta)} \right) \leq \delta. \tag{C.1}$$

We define the event $\mathcal{E}$ as follows $\mathcal{E} = \cap \mathcal{E}_r$, where

$$\mathcal{E}_r := \left\{ \forall y \in \mathcal{Y}(S_r), (\theta_r - \theta_*)^\top y \leq \|y\|_{(\sum_{x \in X_r} xx^\top)^{-1}} \cdot 2\sqrt{\log(|\mathcal{Y}(S_r)|/\delta_r)} \right\} \tag{C.2}$$

Lemma C.1 suggests that $\mathcal{E}_r$ holds with probability at least $1 - \delta_r$, therefore we have $\mathcal{E}$ holds w.p. at least $1 - \sum_r \delta_r \geq 1 - \delta$.

Next we assume $\mathcal{E}$ holds. We show that $1/3\Delta_x \leq \epsilon_x \leq 5/3\Delta_x$. First we have

$$|\epsilon_x^r - \Delta_x| = |\theta_r^\top(x_*^r - x) - \theta_*^\top(x_* - x)|. \tag{C.3}$$

To further bound it, for any $x \in S_r$, we have

$$\begin{aligned}
\theta_r^\top(x_*^r - x) - \theta_*^\top(x_* - x) &= (\theta_r - \theta_*)^\top(x_*^r - x) + \theta_*^\top(x_*^r - x_*) \\
&\leq (\theta_r - \theta_*)^\top(x_*^r - x) \\
&\leq \max_{y \in \mathcal{Y}(S_r)} \|y\|_{(\sum_{i=1}^N x_i x_i^\top)^{-1}} \cdot 2\sqrt{\log(|S_r|^2/\delta)},
\end{aligned} \tag{C.4}$$

where we use Lemma C.1 and the fact that $x_*^r \in S_r$. Next, we further have

$$\begin{aligned}
\theta_r^\top(x_*^r - x) - \theta_*^\top(x_* - x) &= \theta_r^\top(x_*^r - x_*) - (\theta_* - \theta_r)^\top(x_* - x) \\
&\geq -(\theta_* - \theta_r)^\top(x_* - x) \\
&\geq -\max_{y \in \mathcal{Y}(S_r)} \|y\|_{(\sum_{i=1}^N x_i x_i^\top)^{-1}} \cdot 2\sqrt{\log(|S_r|^2/\delta)},
\end{aligned} \tag{C.5}$$

where we use the fact that $\theta_r^\top(x_*^r - x_*) \geq 0$. Therefore, for any $x \in S_r$, we have

$$|\epsilon_x^r - \Delta_x| \leq 4 \max_{y \in \mathcal{Y}(S_r)} \|y\|_{(\sum_{i=1}^N x_i x_i^\top)^{-1}} \cdot \sqrt{\log(|S_r|/\delta)} \leq 2\sqrt{\log(|S_r|^2/\delta)(1+\epsilon)\rho_r/N_r}. \tag{C.6}$$

Then for all $x \in O_r$, we always have

$$\Delta_x \geq \epsilon_x - |\epsilon_x - \Delta_x| \geq 3\sqrt{\log(|S_r|^2/\delta)(1+\epsilon)\rho_r/N_r} \geq 3/2 \cdot |\epsilon_x - \Delta_x|,$$

which means $1/3\Delta_x \leq \epsilon_x \leq 5/3\Delta_x$.

## C.1 PROOF OF CORRECTNESS OF ALGORITHM 2

*Proof of Correctness of Algorithm 2.* Suppose $\mathcal{E}$ holds. Let $x_*^r$ be the $x$ with $\max_x \hat{p}_x^r$. Assume that $x_* \in S_r$. Then we want to prove that $x_* \in S_{r+1}$ with high probability. Actually we have with probability $1 - \delta_r$,

$$\begin{aligned}
\epsilon_{x_*}^r &= \theta_*^\top(x_*^r - x_*) + (\theta_r - \theta_*)^\top(x_*^r - x_*) \\
&\leq (\theta_r - \theta_*)^\top(x_*^r - x_*) \\
&\leq \max_{y \in \mathcal{Y}(S_r)} \|y\|_{(\sum_{x \in X_r} xx^\top)^{-1}} \cdot 2\sqrt{\log(|\mathcal{Y}(S_r)|/\delta_r)} \\
&\leq \max_{y \in \mathcal{Y}(S_r)} \|y\|_{(\sum_{x \in X_r} (\lambda_r^*)_x xx^\top)^{-1}} \cdot 2\sqrt{(1+\epsilon)\log(|\mathcal{Y}(S_r)|/\delta_r)/N_r} \\
&\leq 2\sqrt{\log(|S_r|^2/\delta_r)(1+\epsilon)\rho_r/N_r} \\
&< 5/\sqrt{L_r},
\end{aligned} \tag{C.7}$$

where the first inequality holds since $x_*$ maximizes $\theta_*^\top x$, the second one holds due to Lemma C.1 and the fact that $x_*^r, x_* \in S_r$, the third one holds due to the ROUND function, the fourth one holds since $|\mathcal{Y}(S_r)| \leq |S_r|^2$ and the definition of $\rho_r$, the last one holds due to the definition of $N_r$. Therefore, we know that $x_*$ will not be discarded at round $r$ under $\mathcal{E}$. $\square$

## C.2 Proof of batch complexity of Algorithm 2

We first have the following lemma, which restates Lemma 4.4.

**Lemma C.2.** *For any $L > 0$, we have*

$$L \cdot \rho(\mathcal{Y}(z \in X : \Delta_z \leq 15/\sqrt{L})) \leq 900\psi^*. \tag{C.8}$$

*Proof.* We have

$$
\begin{aligned}
&L \cdot \rho(\mathcal{Y}(X \setminus \{x : \Delta_x > 15/\sqrt{L}\})) \\
&= L \cdot \min_{\lambda \in \Delta^X} \max_{y \in \mathcal{Y}(X \setminus \{x : \Delta_x > 15/\sqrt{L}\})} \|y\|^2_{(\sum_{x \in X} \lambda_x x x^\top)^{-1}} \\
&\leq 4 \cdot L \cdot \min_{\lambda \in \Delta^X} \max_{y \in X \setminus \{x : \Delta_x > 15/\sqrt{L}\}} \|y - x^*\|^2_{(\sum_{x \in X} \lambda_x x x^\top)^{-1}} \\
&\leq 4 \cdot 225 \cdot \min_{\lambda \in \Delta^X} \max_{y \in X \setminus \{x : \Delta_x > 15/\sqrt{L}\}} \frac{\|y - x^*\|^2_{(\sum_{x \in X} \lambda_x x x^\top)^{-1}}}{\Delta_y^2} \\
&\leq 4 \cdot 225 \cdot \min_{\lambda \in \Delta^X} \max_{y \in X} \frac{\|y - x^*\|^2_{(\sum_{x \in X} \lambda_x x x^\top)^{-1}}}{\Delta_y^2} \\
&= 900 \cdot \psi^*,
\end{aligned}
\tag{C.9}
$$

where the first inequality holds since $\max_{a,b \in A} \|a - b\| \leq 2 \max_{a \in A} \|a - c\|$ for some set $A$, some $c \in A$ and some metric $\|\cdot\|$; the second one holds since for any $y \in X \setminus \{x : \Delta_x > 15 \cdot /\sqrt{L}\}$, we have $L \leq 225/\Delta_y^2$. $\qquad\square$

Now we start to prove batch complexity of Algorithm 2.

*Proof of batch complexity of Algorithm 2.* Suppose $\mathcal{E}$ holds. Similar to the MAB setting, we use induction to prove the batch complexity of Algorithm 2. Recall that $T_r$ satisfies that $4^{T_r} \leq L_r < 4^{T_r+1}$ and

$$
\begin{aligned}
L_{r+1} &\leftarrow 4L_r + \frac{\sum_{t=1}^{T_r} 4^t \cdot \rho(\mathcal{Y}(X \setminus \{x \in \cup_{s=1}^r O_s : \epsilon_x > 2^{-t} \cdot 5/3\}))}{\rho(\mathcal{Y}(X \setminus \cup_{s=1}^r O_s))} \\
&\geq 4L_r + \frac{\sum_{t=1}^{T_r} 4^t \cdot \rho(\mathcal{Y}(X \setminus \{x \in \cup_{s=1}^r O_s : \Delta_x > 2^{-t}\}))}{\rho(\mathcal{Y}(X \setminus \cup_{s=1}^r O_s))},
\end{aligned}
\tag{C.10}
$$

where we use the fact that $\epsilon_x \leq 5/3\Delta_x$ under the event. We define the virtual sequence $\bar{L}_r$ as follows: $\bar{T}_r$ satisfies that $\bar{L}_r = 4^{\bar{T}_r}$ and

$$
\hat{L}_{r+1} \leftarrow 4\bar{L}_r + \frac{\sum_{t=1}^{\bar{T}_r} 4^t \cdot \rho(\mathcal{Y}(X \setminus \{x \in U_r : \Delta_x > 15 \cdot 2^{-t}\}))}{\rho(\mathcal{Y}(X \setminus U_r))}, \quad U_r := \{x | \Delta_x > 15/\sqrt{\bar{L}_r}\}.
\tag{C.11}
$$

and we select $\bar{L}_{r+1} = 4^T$ satisfying $4^T \leq \hat{L}_{r+1} < 4^{T+1}$.

Next, we use induction to prove that

$$\bar{L}_r \leq L_r, \tag{C.12}$$
$$U_r \subseteq \cup_{s=1}^r O_s. \tag{C.13}$$

Suppose both (C.12) and (C.13) hold for $1, ..., r$. For $r + 1$, we first prove (C.12) holds.

Since (C.13) holds for $r$, we have $U_r \subseteq \cup_{s=1}^r O_s$. Note that by the definition of $\bar{T}_r$, we have another definition of $U_r$, which is $U_r = \{x | \Delta_x > 15 \cdot 2^{-\bar{T}_r}\}$. Therefore, for all $t \leq \bar{T}_r$, we have

$$\{x \in U_r : \Delta_x > 15 \cdot 2^{-t}\} = \{x : \Delta_x > 15 \cdot 2^{-t}\}. \tag{C.14}$$

Meanwhile, since $U_r \subseteq \cup_{s=1}^r O_s$, we have

$$\{x \in U_r : \Delta_x > 15 \cdot 2^{-t}\} \subseteq \{x \in \cup_{s=1}^r O_s : \Delta_x > 15 \cdot 2^{-t}\} \subseteq \{x : \Delta_x > 15 \cdot 2^{-t}\}. \tag{C.15}$$

Therefore, combining (C.14) and (C.15), we have $\{x \in U_r : \Delta_x > 15 \cdot 2^{-t}\} = \{x \in \cup_{s=1}^r O_s : \Delta_x > 15 \cdot 2^{-t}\}$. Then we have

$$\rho(\mathcal{Y}(X \setminus \{x \in \cup_{s=1}^r O_s : \Delta_x > 15 \cdot 2^{-t}\})) = \rho(\mathcal{Y}(X \setminus \{x \in U_r : \Delta_x > 15 \cdot 2^{-t}\})). \quad \text{(C.16)}$$

Since (C.12) holds for $r$, it suggests that $\bar{T}_r \leq T_r$. Then following (C.10), we have

$$
\begin{aligned}
L_{r+1} &\geq 4L_r + \frac{\sum_{t=1}^{T_r} 4^t \cdot \rho(\mathcal{Y}(X \setminus \{x \in \cup_{s=1}^r O_s : \Delta_x > 15 \cdot 2^{-t}\}))}{\rho(\mathcal{Y}(X \setminus \cup_{s=1}^r O_s))} \\
&\geq 4\bar{L}_r + \frac{\sum_{t=1}^{\bar{T}_r} 4^t \cdot \rho(\mathcal{Y}(X \setminus \{x \in \cup_{s=1}^r O_s : \Delta_x > 15 \cdot 2^{-t}\}))}{\rho(\mathcal{Y}(X \setminus U_r))} \\
&= 4\bar{L}_r + \frac{\sum_{t=1}^{\bar{T}_r} 4^t \cdot \rho(\mathcal{Y}(X \setminus \{x \in U_r : \Delta_x > 15 \cdot 2^{-t}\}))}{\rho(\mathcal{Y}(X \setminus U_r))} \\
&\geq \bar{L}_{r+1}, \quad \text{(C.17)}
\end{aligned}
$$

where the second inequality holds since $U_r \subseteq \cup_{s=1}^r O_s$ and $T_r \geq \bar{T}_r$, the last inequality holds due to the definition of $\bar{L}_{r+1}$. Thus, we prove that (C.12) for $r+1$.

Next we show that (C.13) also holds for $r+1$. Consider $O_{r+1}$, whose definition is

$$O_{r+1} := \{x \in X \setminus \cup_{s=1}^r O_s | \epsilon_x^{r+1} > 5/\sqrt{L_{r+1}}\}. \quad \text{(C.18)}$$

For any $x \in U_{r+1}$, we have $x \in U_{r+1} \Rightarrow \epsilon_x^{r+1} > 5/\sqrt{L_{r+1}}$, which can be derived as follows:

$$x \in U_{r+1} \Leftrightarrow \Delta_x \geq 15/\sqrt{\bar{L}_{r+1}} \Rightarrow \Delta_x \geq 15/\sqrt{L_{r+1}} \Rightarrow \epsilon_x^{r+1} > 5/\sqrt{L_{r+1}}. \quad \text{(C.19)}$$

We now consider any $x \in U_{r+1}$. If $x \in \cup_{s=1}^r O_s$, the induction is done. Otherwise, we have $x \in O_{r+1}$ by definition, which again suggests that $x \in \cup_{s=1}^{r+1} O_s$. Then (C.13) also holds for $r+1$.

Next we only consider the sequence $\bar{L}_r := 4^{\bar{T}_r}$. We consider the sequence

$$H_r := \sum_{t=1}^{\bar{T}_r} 4^t \cdot \rho(\mathcal{Y}(X \setminus \{x \in U_r : \Delta_x > 15 \cdot 2^{-t}\})), \ U_r = \{x : \Delta_x > 15/\sqrt{\bar{L}_r}\}. \quad \text{(C.20)}$$

Based on $H_r$, we have the definition of $\hat{L}_{r+1}$, which is

$$\hat{L}_{r+1} = 4\bar{L}_r + H_r/\rho(\mathcal{Y}(X \setminus U_r)), \ 1/4 \cdot \hat{L}_{r+1} \leq \bar{L}_{r+1} < \hat{L}_{r+1}. \quad \text{(C.21)}$$

First we would like to show that $H_r \leq H_{r+1}$. It is obivious since $\bar{T}_{r+1} > \bar{T}_r$ due to the fact that $\bar{L}_{r+1} \geq 4\bar{L}_r$.

Next we would like to show that $4 \cdot \rho(\mathcal{Y}^*(X)) \leq H_r \leq 900 \log_2(1/\Delta_2)\psi^*$. For the left hand side, note that

$$H_r \geq H_1 = 4 \cdot \rho(\mathcal{Y}(X)) \geq 4 \cdot \rho(\mathcal{Y}^*(X)). \quad \text{(C.22)}$$

For any $t \leq \bar{T}_r$ we have

$$4^t \cdot \rho(\mathcal{Y}(X \setminus \{x \in U_r : \Delta_x > 15 \cdot 2^{-t}\})) = 4^t \cdot \rho(\mathcal{Y}(X \setminus \{x : \Delta_x > 15 \cdot 2^{-t}\})) \leq 900 \cdot \psi^*,$$

where the inequality holds due to (C.8). Therefore, $H_r \leq 900\psi^*(\bar{T}_r) \leq 900 \log_2(1/\Delta_2)\psi^*$ where we use the vacous bound that $\bar{T}_r \leq \log_2(1/\Delta_2)$.

Finally, we show that for any $r$ satisfying when $U_r = U_{r+1}$, we have $H_{r+1} \geq 5/4 \cdot H_r$. We have

$$
\begin{aligned}
H_{r+1} &= \sum_{t=1}^{\bar{T}_{r+1}} 4^t \cdot \rho(\mathcal{Y}(X \setminus \{x \in U_{r+1} : \Delta_x > 15 \cdot 2^{-t}\})) \\
&= \sum_{t=1}^{\bar{T}_{r+1}} 4^t \cdot \rho(\mathcal{Y}(X \setminus \{x \in U_r : \Delta_x > 15 \cdot 2^{-t}\}))
\end{aligned}
$$

$$\geq \bar{L}_{r+1} \cdot \rho(\mathcal{Y}(X \setminus U_r)) + \sum_{t=1}^{\bar{T}_r} 4^t \cdot \rho(\mathcal{Y}(X \setminus \{x \in U_r : \Delta_x > 15 \cdot 2^{-t}\}))$$

$$\geq \frac{1}{4} \cdot \sum_{t=1}^{\bar{T}_r} 4^t \cdot \rho(\mathcal{Y}(X \setminus \{x \in U_r : \Delta_x > 15 \cdot 2^{-t}\}))$$

$$+ \sum_{t=1}^{\bar{T}_r} 4^t \cdot \rho(\mathcal{Y}(X \setminus \{x \in U_r : \Delta_x > 15 \cdot 2^{-t}\}))$$

$$\geq \frac{5}{4} \cdot H_r, \tag{C.23}$$

where the first inequality holds since $\bar{L}_{r+1} = 4^{\bar{T}_{r+1}}$, the second one holds due to the definition of $\bar{L}_{r+1}$, the last one holds trivially. Therefore, it is easy to bound $R$ as

$$R \leq \alpha + \log_{5/4}(900 \frac{\log_2(1/\Delta_2)\psi^*}{\rho(\mathcal{Y}^*(X))}), \tag{C.24}$$

where $\alpha$ is the number of different $U_r$. $\qquad\square$

### C.3 Proof of Sample Complexity of Algorithm 2

To prove the sample complexity, we first show the following lemma holds, which includes several claims.

**Lemma C.3.** *Suppose $\mathcal{E}$ holds. Then for any $r$, we have the following claim:*

$$S_{r+1} = (X \setminus \cup_{s=1}^{r} O_s) \subseteq \{z \in X : \Delta_z \leq 15/\sqrt{L_r}\}. \tag{C.25}$$

*Furthermore, for any $t \leq T_r$, we have*

$$(X \setminus \{x \in \cup_{s=1}^{r} O_s : \Delta_x > 5 \cdot 2^{-t}\}) \subseteq \{z \in X : \Delta_z \leq 15 \cdot 2^{-t}\}. \tag{C.26}$$

*Proof.* (C.25) holds since $z \notin \cup_{s=1}^{r} O_s \Rightarrow \epsilon_z^r \leq 5/\sqrt{L_r}$, which suggests that $\Delta_z \leq 15/\sqrt{L_r}$ by the fact that $\epsilon_z^r > \frac{1}{3}\Delta_z$. To show (C.26), we fix $r$ and $t \leq T_r$, and note that for any $z \in X \setminus \{x \in \cup_{s=1}^{r} O_s : \Delta_x > 5 \cdot 2^{-t}\}$, we have either

1. $z \notin \cup_{s=1}^{r} O_s$, or

2. $z \in \cup_{s=1}^{r} O_s$ and $\Delta_z \leq 5 \cdot 2^{-t}$.

For the first case, we have $\Delta_z \leq 15/\sqrt{L_r}$ by (C.25). Using the definition of $T_r$ ($4^{T_r} \leq L_r < 4^{T_r+1}$), we have $L_r \geq 4^t$, which further suggests that $\Delta_z \leq 15 \cdot 2^{-t}$. For the second case, we always have $\Delta_z \leq 5 \cdot 2^{-t} \leq 15 \cdot 2^{-t}$. Therefore, (C.26) holds. $\qquad\square$

Next we start to prove the sample complexity of Algorithm 2.

*Proof of sample complexity of Algorithm 2.* Suppose $\mathcal{E}$ holds. Recall the definition of $L_{r+1}$, we have

$$L_{r+1} = 4L_r + \frac{\sum_{t=1}^{T_r} 4^t \cdot \rho(\mathcal{Y}(X \setminus \{x \in \cup_{s=1}^{r} O_s : \epsilon_x > 2^{-t} \cdot 5/3\}))}{\rho(\mathcal{Y}(X \setminus \cup_{s=1}^{r} O_s))}$$

$$\leq 4L_r + \frac{\sum_{t=1}^{T_r} 4^t \cdot \rho(\mathcal{Y}(X \setminus \{x \in \cup_{s=1}^{r} O_s : \Delta_x > 5 \cdot 2^{-t}\}))}{\rho(\mathcal{Y}(X \setminus \cup_{s=1}^{r} O_s))},$$

$$\leq 4L_r + \frac{\sum_{t=1}^{T_r} 4^t \cdot \rho(\mathcal{Y}(\{z \in X : \Delta_z \leq 15 \cdot 2^{-t}\}))}{\rho(\mathcal{Y}(X \setminus \cup_{s=1}^{r} O_s))}, \tag{C.27}$$

where for the first inequality, we use the fact $\epsilon_x > \frac{1}{3}\Delta_x$, and the second one holds due to (C.26). Now, we start to bound $N_{r+1}$, which is

$$N_{r+1} = 4 \max\{2 \log(|S_{r+1}|^2/\delta_{r+1})\rho_{r+1}L_{r+1}, d\}$$

$$\leq 4d + 16 \log(|X|/\delta_{r+1}) \rho_{r+1} L_{r+1}$$

$$= 4d + 16 \log(|X|/\delta_{r+1}) \rho(\mathcal{Y}(X \setminus \cup_{s=1}^{r} O_s))$$

$$\cdot \left( 4L_r + \frac{\sum_{t=1}^{T_r} 4^t \cdot \rho(\mathcal{Y}(\{z \in X : \Delta_z \leq 15 \cdot 2^{-t}\}))}{\rho(\mathcal{Y}(X \setminus \cup_{s=1}^{r} O_s))} \right)$$

$$= 4d + 16 \log(|X|/\delta_{r+1})$$

$$\cdot \left( 4L_r \rho(\mathcal{Y}(X \setminus \cup_{s=1}^{r} O_s)) + \sum_{t=1}^{T_r} 4^t \cdot \rho(\mathcal{Y}(\{z \in X : \Delta_z \leq 15 \cdot 2^{-t}\})) \right)$$

$$\leq 4d + 16 \log(|X|/\delta_{r+1})$$

$$\cdot \left( 4L_r \rho(\mathcal{Y}(\{z \in X : \Delta_z \leq 15/\sqrt{L_r}\})) + \sum_{t=1}^{T_r} 4^t \cdot \rho(\mathcal{Y}(\{z \in X : \Delta_z \leq 15 \cdot 2^{-t}\})) \right)$$

$$\leq 4d + 16 \log(|X|/\delta_{r+1}) \left( 3600\psi^* + T_r \cdot 900\psi^* \right), \tag{C.28}$$

where the second inequality holds due to (C.25) and the last one holds due to (C.8). Therefore, by (C.28), let $B$ denote the total number of batches which is always bounded by $\log(1/\Delta_2)$, we have

$$N = 4\rho(\mathcal{Y}(X)) + \sum_{r=1}^{B-1} N_{r+1} \leq O(\psi^* \log^2(1/\Delta_2) \log(|X|/\delta) + \log(1/\Delta_2)d),$$

where we use the fact that $\rho(\mathcal{Y}(X)) \leq d$, which is the standard property of design $\rho$ (see Section 21.1 in Lattimore & Szepesvári (2020)).

$\square$