# OpenReview forum: "Breaking the $\log(1/\Delta_2)$ Barrier: Better Batched Best Arm Identification with Adaptive Grids"
_ICLR.cc/2025/Conference — ICLR 2025 Poster_

### Official Review · Reviewer_2sNi · 2024-10-31

**Soundness:** 3
**Presentation:** 2
**Contribution:** 2
**Rating:** 6
**Confidence:** 3

**Summary:**

The authors conducted a study of the batched BAI problem in both MAB and linear bandit settings, and improved the existing bounds. They proposed IS-SE and IS-RAGE, which stem from the previous algorithms and both have good performance in some cases (outperformed the O(1/Delta) bound). The experiments also corroborated these results.

**Strengths:**

The authors proposed a new algorithm that, in some ways, improves the original result in MAB and linear bandits setting. The theoretical result is solid. The experiment was compared with the SE algorithm comprehensively.

**Weaknesses:**

The authors need to compare with more algorithms beyond SE. The novelty of the proposed algorithm is not clearly stated. Also, why is the proposed solution better?

**Questions:**

The examples mention that: in certain cases, IS-SE can have fewer batches than log(1/Delta). Why is this the case and when would this happen?
- In this case, do other algorithms also need log(1/Delta) batch, or just the same as IS-SE？
- Does it truly imply O(1) batches ( since SE still need log(1/Delta))? This is not so clear.

Post-rebuttal: the authors have provided responses that help answer these questions.

---

> ### Author Response · Authors · 2024-11-14
> **Clarification on Reviewer 2sNi's Comments**
>
> Thank you for your insightful comments. We have a few clarifying questions regarding your concerns and would appreciate your feedback.
>
> 1. **In certain cases, IS-SE can have fewer batches than $\log(1/\Delta)$. Why is this the case, and when would this happen?**
>
>    Could you clarify your concern here? Are you questioning why IS-SE can achieve a batch complexity lower than $\log(1/\Delta_2)$, or are you uncertain about the calculation of the batch complexity for IS-SE? Additionally, regarding the question of "when would this happen," would you like us to provide more examples where IS-SE’s batch complexity outperforms $\log(1/\Delta_2)$?
>
> 2. **Does this truly imply $\mathcal{O}(1)$ batches (since SE still needs $\log(1/\Delta_2)$)? This is not entirely clear.**
>
>    Are you seeking further details on the calculation of IS-SE's batch complexity as it pertains to Examples 1-3?
>
> Thank you, and we look forward to your guidance on these points.

---

> > ### Comment · Reviewer_2sNi · 2024-11-15
> >
> > For 1, could you provide more details about why it can achieve lower batch complexity? Yes, more examples would help.
> >
> > For 2, can you give more details regarding the calculation of example 3?

---

> > > ### Author Response · Authors · 2024-11-22
> > > **Response to Reviewer 2sNi**
> > >
> > > Thanks for your comments.
> > >
> > > **Q1&Q2:** Could you provide more details about why IS-SE can achieve lower batch complexity? Can you give more details regarding the calculation of example 3?
> > >
> > > **A1&A2:** Sure. We will use Ex. 2 as an example to demonstrate why our algorithms have advantage over SE. We will also give the detailed calculation for Ex. 3.
> > >
> > > Recall the definition of $R_I$. We start with $\bar{L}\_0=1$ and $U\_{0}=\emptyset$, and recursively define for $r=1,2,\cdots$ the quantities
> > > $\bar{L}\_{r}=4\bar{L}\_{r-1} +\frac{1}{n-|U\_{r-1}|}\sum\_{j\in U\_{r-1}} \frac{1}{\Delta_j^2},$ and $U\_{r}=\{j: \Delta\_j\geq  \sqrt{\frac{1}{\bar{L}\_r}}\}.$
> > >
> > > We stop when $U\_r = [n]\backslash \{\text{best arm}\}$. Let $R_I$ be the value of $r$ when we stop.  (We have set $C=1$ here for simplicty in the calculation. This choice does not affect the asymptotic complexity of $R_I$)
> > >
> > > In Ex. 2, the best arm has mean $\frac{1}{2}$; $\big(n-\log_2 n+1\big)$ arms have mean $\left(\frac{1}{2}-\frac{1}{2}\right)$; $1$ arm has mean $\left(\frac{1}{2}-\frac{1}{4}\right)$; $1$ arm has mean $\left(\frac{1}{2}-\frac{1}{8}\right)$; $\ldots$, and $1$ arm has mean $\left(\frac{1}{2}-\frac{1}{\sqrt{n}}\right)$.
> > >
> > > We have $\bar{L}\_{1}=4$, $U\_{1}=\{ \text{arms with gap at least } 1/2 \}$.
> > >
> > > $$\bar{L}\_{2}= 4 \cdot \bar{L}\_{1} + \frac{1}{\log_2 n - 1}\cdot (n-\log_2n+1) \cdot 4 = \frac{4n}{\log n} + o\left(\frac{n}{\log n}\right),$$
> > >
> > > and $U_2= [n] \backslash \{\text{best arm}\}$. Therefore, $R_I = 2$. On the other hand, the batch complexity of SE is $\log (1/\Delta_2) = \log\sqrt{n} = \Theta(\log n)$.
> > >
> > >
> > >
> > >
> > > In Ex. 3, for simplicity, let $x = 4^r$ for some $r\geq 1$, and the best arm has mean $\frac{1}{2}$; $x$ arms have gap $\frac{1}{2}$; $\frac{x}{4}$ arms have gap $\frac{1}{4},\cdots$; $4^j$ arms have gap $\frac{1}{2\cdot 2^{r-j}}$ for $j = 0,\dots, r$. We also have $n = \frac{4^{r+1}+2}{3}$.
> > >
> > > We next aim to give $\bar L_j$ both upper bounds and lower bounds.
> > > For the lower bound, it is straightforward to see that $\bar L_j \geq 4^j$.
> > > For the upper bound, we use induction to prove it.
> > >
> > > -Suppose that for some $j$, there exists some $0<i\leq r$ such that $4^{i-1}\leq \bar L_{j-1}<4^i$. Then by the definition of $U_{j-1}$, we know that $U_{j-1}\subset \{\text{arms with gap at least }\frac{1}{2^i}\}$.
> > >
> > > -According to the definition of Ex. 3, we know that $n-|U_{j-1}| \geq |\{\text{arms with gap smaller than }\frac{1}{2^i}\}| = 1+1+\cdots + 4^{r-i} = \frac{4^{r-i+1}+2}{3}$, and $\sum_{t \in U_{j-1}}\frac{1}{\Delta_t^2} \leq \sum_{t: \Delta_t=\frac{1}{2}, \frac{1}{4},\dots, \frac{1}{2^i}}\frac{1}{\Delta_t^2}=\frac{x}{(\frac{1}{2})^2} + \frac{\frac{x}{4}}{(\frac{1}{4})^2} + \dots =4ix=4^{r+1}i$.
> > >
> > > -Therefore, by the definition of $\bar L\_{j}=4\bar{L}\_{j-1} +\frac{1}{n-|U\_{j-1}|}\sum_{t\in U\_{j-1}} \frac{1}{\Delta_t^2}$, we have $\bar L\_{j}\leq 4^{i+1} +\frac{1}{\frac{4^{r-i+1}+2}{3}}4^{r+1}i \leq 7i\cdot 4^i \leq 7r\bar L_{j-1}$.
> > >
> > > -Therefore, by induction, we have $\bar L\_j \leq (7r)^{j-1}$.
> > >
> > > Finally, since $\bar L_j\geq 4^j$, we have $R_I \leq r = O(\log n)$, since $\bar L_j \leq (7r)^{j-1}$, we have $R_I\geq \log_{7r}(4^{r+1}) = \frac{r+1}{\log(7r)} = \Omega(\frac{\log n}{\log \log n})$. Omitting $\log\log n$, we have $R_I = \Theta (\log n)$.

---

> > > > ### Comment · Reviewer_2sNi · 2024-11-24
> > > >
> > > > Thanks. A quick follow-up: does running the SE algorithm in EX.2 reach log(1/delta-2)?

---

> > > > > ### Author Response · Authors · 2024-11-24
> > > > >
> > > > > Yes, in all the three examples (including Ex. 2) discussed in Section 1.1, the SE algorithm needs $\log(1/\Delta_2)$ batches.  Intuitively, in the $r$-th round, SE eliminates (and only eliminates) all arms $i$ with mean gaps $\Delta_i \ge 1/2^r$.  Therefore, it needs $\log(1/\Delta_2)$ batches to eliminate the arm with mean gap $\Delta_2$.

---

> > > > > > ### Comment · Reviewer_2sNi · 2024-11-24
> > > > > >
> > > > > > Thanks. I have no further questions. I've increased the score to reflect my updated evaluation of this paper.

---

> > > > > > > ### Author Response · Authors · 2024-11-24
> > > > > > >
> > > > > > > Thank you for your positive feedback!

---

### Official Review · Reviewer_GT9n · 2024-11-03

**Soundness:** 2
**Presentation:** 2
**Contribution:** 2
**Rating:** 3
**Confidence:** 4

**Summary:**

This paper proposes two algorithms, instance-sensitive successive elimination and RAGE, where the batch complexities can outperform the $\log(1/\Delta_2)$ bound.

**Strengths:**

(1) Theoretically, the paper demonstrates that the proposed algorithms achieve near-optimal sample complexity while breaking the barrier for batch complexity.

(2) An instance-sensitive quantity, $R_I$, is introduced.

**Weaknesses:**

(1) A significant issue is that the two algorithms lack empirical support, making it challenging to convince readers of their practical value.

(2) Breaking the $\log(1/\Delta_2)$ barrier appears to be conditional, depending on the number of arms with nearly optimal means being relatively limited compared to the total number of arms. In such cases, the allocation may rapidly focus on only a few arms. I am uncertain whether these algorithms offer any advantage over the original SE under these conditions.

(3) The issue of a small gap $\Delta_2$ is more serious in the fixed-budget setting and has been observed in Bayesian Best Arm Identification (BAI). For example, the paper Exploration for Fixed-Budget Bayesian Best Arm Identification (arXiv: 2408.04869) discusses this. More related work should be considered.

(4) Some parts of the paper are difficult to read, possibly due to the theoretical depth. For instances, it’s challenging to understand $R_1$ from Definition 1.1 across the three examples in Figure 1. $\Delta_2$ is undefined, though it appears to represent the gap between the best and second-best arms.

**Questions:**

See above

---

> ### Author Response · Authors · 2024-11-22
> **Response to Reviewer GT9n**
>
> Thank you for your comments.
>
> **Q1:** A significant issue is that the two algorithms lack empirical support, making it challenging to convince readers of their practical value.
>
> **A1:** Thank you for your feedback. We would like to emphasize that our original submission included experimental results for the MAB setting in Section 3.3, demonstrating the advantages of IS-SE in terms of the sample complexity-batch complexity trade-off. To strengthen the empirical support, we have added experiments in our revision for both linear bandits and real-world datasets in Section 5 of the new version. The results consistently show that our frameworks improve the sample complexity-batch complexity trade-off, providing further evidence of their practical value.
>
>
> **Q2:**  Breaking the $\log (1/\Delta_2)$ barrier appears to be conditional, depending on the number of arms with nearly optimal means being relatively limited compared to the total number of arms. In such cases, the allocation may rapidly focus on only a few arms. I am uncertain whether these algorithms offer any advantage over the original SE under these conditions.
>
> **A2:** The reviewer is correct that “breaking the $\log (1/\Delta_2)$ barrier appears to be conditional.” Indeed, as mentioned in the abstract of our paper, the batch complexities of our algorithms depend on the input instance, particularly the mean gaps of the suboptimal arms. This instance dependency is precisely why we describe our batch complexity as instance-sensitive. Compared to SE (which achieves a $\log (1/\Delta_2)$ batch bound), the advantage of our algorithm lies in its continuous estimation of the instance sample complexity $H_I$, which is then used to allocate the sample budget for the next round.
>
> However, we are unclear about what the reviewer meant by “whether these algorithms offer any advantage over the original SE under these conditions.”  If the reviewer wanted to see more explanations on the examples that we provide in our submission, we would like to use Ex. 2 as an example to demonstrate why our algorithms have advantage over SE.
>
> Recall the definition of $R_I$. We start with $\bar{L}\_0=1$ and $U\_{0}=\emptyset$, and recursively define for $r=1,2,\cdots$ the quantities
>
> $\bar{L}\_{r}=4\bar{L}\_{r-1} +\frac{1}{n-|U\_{r-1}|}\sum\_{j\in U\_{r-1}} \frac{1}{\Delta\_j^2}$  and  $U\_{r}=\{j: \Delta\_j\geq  \sqrt{\frac{1}{\bar{L}\_r}}  \}.$
>
> We stop when $U\_r = [n]\backslash \{\text{best arm}\}$. Let $R_I$ be the value of $r$ when we stop.  (We have set $C=1$ here for simplicty in the calculation. This choice does not affect the asymptotic complexity of $R_I$)
>
>
>
>
> In Ex. 2, the best arm has mean $\frac{1}{2}$; $(n-\log_2 n+1)$ arms have mean $(\frac{1}{2}-\frac{1}{2})$; $1$ arm has mean $(\frac{1}{2}-\frac{1}{4})$; $1$ arm has mean $(\frac{1}{2}-\frac{1}{8})$; $\ldots$, and $1$ arm has mean $(\frac{1}{2}-\frac{1}{\sqrt{n}})$.
>
> We have $\bar{L}\_{1}=4$, $U\_{1}=\{ \text{arms with gap at least } 1/2 \}$. Then we have
>
> $$\bar{L}\_{2}= 4 \cdot \bar{L}\_{1} + \frac{1}{\log_2 n - 1}\cdot (n-\log_2n+1) \cdot 4 = \frac{4n}{\log n} + o(\frac{n}{\log n}),$$ and $U_2= [n] \backslash \{\text{best arm}\}$. Therefore, $R_I = 2$. On the other hand, the batch complexity of SE is $\log (1/\Delta_2) = \log\sqrt{n} = \Theta(\log n)$.
>
>
> **Q3:** The issue of a small gap $\Delta_2$ is more serious in the fixed-budget setting and has been observed in Bayesian Best Arm Identification (BAI). For example, the paper Exploration for Fixed-Budget Bayesian Best Arm Identification (arXiv: 2408.04869) discusses this. More related work should be considered.
>
> **A3:**  We thank the reviewer for providing a reference. However, we would like to clarify that our paper focuses on batch bandits for the best-arm identification task in a fixed-confidence setting. In contrast, the paper “Fixed-Budget Bayesian Best Arm Identification” addresses the fixed-budget best-arm identification problem in a Bayesian setting and does not consider batch complexity. As such, we believe this paper is not directly relevant to our work in terms of either techniques or topics. Concrete pointers to the missing works would be very helpful, but we will certainly conduct a more thorough review of the relevant literature.

---

> > ### Author Response · Authors · 2024-11-22
> >
> > **Q4:** Some parts of the paper are difficult to read, possibly due to the theoretical depth. For instances, it’s challenging to understand $R_{1}$ from Definition 1.1 across the three examples in Figure 1.
> >
> > **A4:**
> > While the recursive definition of $R_I$ may look complex, we believe the fundamental concept is clearly explained in our paper. The batch complexity of SE is a degenerated case of our definition of $R_I$ by fixing $\bar{L}\_{r} = 4^r$ (see (1.2)). In our algorithm, we added a second term $\frac{1}{n-|U\_{r-1}|} \sum\_{j \in U\_{r-1}}\frac{1}{\Delta_j^2}$ to $\bar{L}\_{r}$, which can be seen as a conservative estimation of the total instance sample complexity of the input.
> >
> > To clarify how suboptimal arms are eliminated, we include Figure 1. For IS-SE, in Ex. 1, all suboptimal arms—except the arm with a mean of $(\frac{1}{2} - \frac{1}{\sqrt{n}})$ are eliminated in the first batch and the second best arm is eliminated in the second batch. Consequently, the process requires only $2$ batches.  Similarly, in Ex. 2, the $\big(n-\log_2 n+1\big)$ arms with mean $\left(\frac{1}{2}-\frac{1}{2}\right)$ will be eliminated in the first batch and all other sub-optimal arms will be eliminated in the second batch. Therefore, the batch complexity of IS-SE is again $2$. While for SE, in both Ex. 1 and Ex. 2, it requires one batch to "visit" each arm level $i$ (i.e., with mean $\frac{1}{2} - \frac{1}{2^i}$), resulting in a total of $\log_2(1/\Delta_2)$ batches.
> >
> > In Ex. 3, both IS-SE and SE need to visit each arm level $i$. Therefore, both algorithms need $\Theta(\log_2(1/\Delta_2))$ batches.
> >
> > If the reviewer thought that the figure is confusing, we would consider removing it and add more word explanations to the three examples instead.
> >
> >
> > **Q5** $\Delta_2$ is undefined, though it appears to represent the gap between the best and second-best arms.
> >
> > **A5:** $\Delta_2$ is defined in Line 57-58. We have added this definition to the abstract of this paper.

---

> > > ### Author Response · Authors · 2024-11-25
> > >
> > > Thank you for your valuable comments. With the ICLR rebuttal phase deadline approaching, we would greatly appreciate any additional feedback or concerns you may have.

---

### Official Review · Reviewer_eBKQ · 2024-11-03

**Soundness:** 2
**Presentation:** 3
**Contribution:** 2
**Rating:** 6
**Confidence:** 4

**Summary:**

The paper considers the problem of batched best-arm identification with a goal of minimizing the batch complexity, which could be an issue for fixed-confidence best-arm identification depending on the minimal gap between the best arm and the second-best arm. The contributions include new algorithms for both k-armed bandit and linear bandit settings as extensions of classic algorithms.

**Strengths:**

- The first algorithm reduces the batch complexity dependence on the minimal gap between the best arm and the second-best arm.
- The paper is well-written and very easy to understand.

**Weaknesses:**

- There are some improvements in the experiments, showing roughly a 2x gain, but it's hard to call them impressive, as the experiments are limited in scale and only cover simple problem instances. Additionally, there are no experiments on the linear bandit setting.
- The contribution mainly reduces a logarithmic factor in batch complexity by transferring it to sample complexity. However, it’s difficult to see an absolute impact on the field.

**Questions:**

The RAGE algorithm, which serves as the building block algorithm in this paper, has publicly available code. Since the proposed algorithm is an extension of it, I wonder if the authors could reproduce some results to demonstrate the improvements.

---

> ### Author Response · Authors · 2024-11-22
> **Response to Reviewer eBKQ**
>
> Thank you for your comments!
>
> **Q1:** There are some improvements in the experiments, showing roughly a 2x gain, but it's hard to call them impressive, as the experiments are limited in scale and only cover simple problem instances. Additionally, there are no experiments on the linear bandit setting.
>
> **A1:** Thank you for your valuable feedback. We understand your concerns regarding the scale and scope of the experiments. First, we would like to highlight that the primary contribution of our work lies in introducing and studying the concept of instance-dependent batch complexity for the first time. Through our theoretical analysis, we propose a refined instance-dependent batch complexity measure, $R_I$, which suggests that significant improvements can be achieved for certain bandit instances. As such, we feel it may not be entirely appropriate to evaluate the quality of our work solely based on the experimental setup.
>
> Regarding the performance gains of our proposed algorithms, we believe the observed improvements in the sample complexity-batch complexity trade-off are indeed meaningful and significant. To address your concerns about the experimental settings, we have expanded the scope of our evaluation to include linear bandit settings and real-world datasets, such as Movielens25M. These new experiments introduce more realistic and challenging scenarios, further validating the effectiveness of our proposed algorithms (IS-SE, IS-RAGE) compared to the baselines (SE, RAGE). We hope these additions provide a more comprehensive view of our contributions and results.
>
>
> **Q2:** The contribution mainly reduces a logarithmic factor in batch complexity by transferring it to sample complexity. However, it’s difficult to see an absolute impact on the field.
>
> **A2:** We believe this is a misunderstanding. To clarify, we present the following two points.
>
> 1: We would like to refer the reviewer to Remark 3.4, Lines 282-288. The previous work (SE) cannot achieve a batch complexity $O(\log (1/\Delta_2))$ by simply increasing its sample complexity by logarithmic factors.
>
> 2: The primary objective of this paper is to show that an $\log(1/\Delta_2)$ batch complexity is *not* instance-optimal and to propose an algorithm that improves upon this batch complexity. Regarding sample complexity, our goal is to achieve a *nearly* optimal solution, up to logarithmic factors. This approach aligns with the methodology of many previous works on batched bandits. For instance, in [1] and [2], the proposed algorithms achieve regret or sample complexity that matches the optimal bounds within logarithmic factors.
>
> [1]: Zijun Gao, Yanjun Han, Zhimei Ren, and Zhengqing Zhou. Batched multi-armed bandits problem. In NeurIPS, 2019.
>
> [2]: Chao Tao, Qin Zhang, and Yuan Zhou. Collaborative learning with limited interaction: Tight bounds for distributed exploration in multi-armed bandits. In FOCS, pp. 126–146. IEEE, 2019.
>
>
>
> Our work shows that for the best-arm identification task, near-optimal sample complexity can be achieved with a batch complexity lower than $\log(1/\Delta_2)$ for many input instances. In some cases, the batch complexity can even be constant. To the best of our knowledge, this instance-sensitive batch complexity is the *first* result of its kind in the bandit literature.
>
>
> **Q3:** Experiments in terms of RAGE for linear bandits.
>
> **A3:**  In our revision, we have included an empirical comparison between RAGE and IS-RAGE in Figure 2.d to Figure 2.f, Section 5, focusing on examples $B_1(n)$ to $B_3(n)$. The results demonstrate that IS-RAGE offers a better sample complexity-batch complexity trade-off compared to RAGE. However, the magnitude of improvement depends on the specific problem instances, similar to the comparisons we conducted for the MAB setting. For example, in instances $B_1(n)$ and $B_2(n)$, the improvements are clear and significant, while for instance $B_3(n)$, the improvement is more modest. We hope this provides a clearer understanding of the advantages of our proposed algorithm.

---

> > ### Comment · Reviewer_eBKQ · 2024-11-24
> >
> > Thank you for the response, especially for adding the experiments on RAGE. I have increased my rating.

---

> > > ### Author Response · Authors · 2024-11-24
> > >
> > > Thanks for your positive feedback!

---

### Official Review · Reviewer_AcUK · 2024-11-04

**Soundness:** 4
**Presentation:** 3
**Contribution:** 4
**Rating:** 8
**Confidence:** 4

**Summary:**

This paper studies the classical best-arm-identification (BAI) problem in the batched setting. The batched setting means that the agent has only limited number of batches. In each batch the agents can arrange which arms and how many pulls for each arm and only receive the results of the pulls after this batch. In this setting, the classical BAI algorithms are not viable.

The authors propose new batched BAI algorithms with adaptive batch sizes (the number of arms pulled in each batch), and achieve a sample complexity of \tilde{O}(\sum_i 1/\Delta_i^2), which is the first to break the bound of \tilde{O}(n/\Delta_2^2), where Delta_2 is the largest reward gap between the optimal and sub-optimal arms.

The authors also extend the batched BAI problems to the linear bandit scenarios and achieved similar results.

**Strengths:**

First, this paper studies a pretty interesting problem. Batched BAI is a very practical setting in industry. For instance for parameter tuning, people will not wait for the last model to complete training before starting another one, given each model training takes days. Batched BAI is a more practical setting for parameter tuning. However, there have not been enough works in this area. Hence, I think this paper is studying an impactful problem.

Second, the results in this paper are good. Ignoring the logarithmic parts, the algorithms have achieved the same sample complexity as the best of BAI. Although I guess ultimately, we should be able to achieve n\log(\delta^-1) sample complexity as classical BAI someday, the n\log(n)\log(n/\delta) is still good enough at this stage. The authors further extend the works to linear bandits.

Third, the algorithms have some novelties. The algorithm uses precise allocations of the batch sizes in order to keep both sample complexity and batch complexity low.

**Weaknesses:**

Some numerical comparisons will strengthen this paper, and also illustrate the improvements of the proposed algorithms.

Some minor suggestions about the writing. 1. Better not use \Delta_2 in abstract while not explaining it. It may cause confusion to some readers. 2. Present more details about the works in CL area, as they share a lot of similarities to the problem in this paper. 3. In the introduction, before using heavy math notations, try to use more natural words to explain what BAI is and what batched BAI is, which can help readers understand what this paper is doing.

**Questions:**

Any study on the lower bound on sample complexity given the lower bound batch complexity?

---

> ### Author Response · Authors · 2024-11-22
> **Response to Reviewer AcUK**
>
> Thanks for your positive feedback!
>
> **Q1:** Some numerical comparisons will strengthen this paper, and also illustrate the improvements of the proposed algorithms.
>
> **A1:** In response to your suggestion, we have expanded our experimental evaluation. In addition to the previously conducted experiments for multi-armed bandits (MAB), we have now included experiments on linear bandits as well as a real-world dataset, Movielens25M. The results demonstrate that our proposed algorithms (IS-SE, IS-RAGE) outperform the baseline methods (SE, RAGE) in terms of the trade-off between sample complexity and batch complexity.
>
> **Q2:** Minor writing suggestions.
>
> **A2:** Thanks for your suggestion. We have added a footnote for $\Delta_2$ to demonstrate its definition, and we also added more discussions about CL during the related work section.
>
> **Q3:** Any study on the lower bound on sample complexity given the lower bound batch complexity?
>
> **A3:** While we believe our sample complexity upper bound is tight up to logarithmic factors, proving a corresponding lower bound poses significant challenges. Existing techniques cannot be directly applied in this setting. We think that the proof would share similarities with the lower bound for best-arm identification (BAI) in the collaborative learning model, as presented in [1]. We would like to mention that the proof in [1] is already highly involved, and extending it to account for both sample and batch/round instance-sensitivity would be even more complex. We leave this as a direction for future work.
>
> [1]: Chao Tao, Qin Zhang, and Yuan Zhou. Collaborative learning with limited interaction: Tight bounds for distributed exploration in multi-armed bandits. In FOCS, pp. 126–146. IEEE, 2019.

---

### Author Response · Authors · 2024-11-22
**Summary of the revision**

Thank you for your valuable feedback. We have revised our draft accordingly. Specifically, we have added experiments comparing RAGE and IS-RAGE in the linear bandit setting, as well as real-world experiments using the Movielens25M dataset. These updates are detailed in Section 5 of the revised version, with changes highlighted in red for clarity. We look forward to hearing your thoughts on the revisions.

---

### Meta-Review · Area_Chair_TSpu · 2024-12-21

**Metareview:**

The paper tackles the practical and impactful problem of batched best-arm identification (BAI), a setting underexplored in prior work yet highly relevant for industrial applications such as parameter tuning, where overlapping model training schedules are common. The results are strong, achieving sample complexity comparable to state-of-the-art BAI approaches, with minor logarithmic overheads and extending the framework to linear bandits. The proposed algorithms are novel in their precise allocation of batch sizes, effectively balancing sample complexity and batch complexity, showcasing innovative contributions to the batched BAI problem.

Considering the above novelties, I recommend acceptance of the paper once the authors incorporate all the reviewer's feedback in the final version.

**Additional Comments On Reviewer Discussion:**

Added experiments and updated the draft. Please refer to the Meta Review.

---

### Decision · Program_Chairs · 2025-01-22

Accept (Poster)